# Ligand response of guanidine-IV riboswitch at single-molecule level

**Lingzhi Gao, Dian Chen, Yu Liu***

State Key Laboratory of Microbial Metabolism, School of Life Sciences and Biotechnology, Shanghai Jiao Tong University, Shanghai, China

## eLife assessment

This study presents **valuable** findings on the ligand- and ion-dependent structural dynamics of a transcriptional riboswitch. The single-molecule data presented are **solid** and prompts intriguing hypotheses and models, which will undoubtedly stimulate future structural analyses. These findings are of considerable interest to biochemists and biophysicists engaged in the study of RNA structure and riboswitch mechanisms.

**Abstract** Riboswitches represent a class of non-coding RNA that possess the unique ability to specifically bind ligands and, in response, regulate gene expression. A recent report unveiled a type of riboswitch, known as the guanidine-IV riboswitch, which responds to guanidine levels to regulate downstream genetic transcription. However, the precise molecular mechanism through which the riboswitch senses its target ligand and undergoes conformational changes remain elusive. This gap in understanding has impeded the potential applications of this riboswitch. To bridge this knowledge gap, our study investigated the conformational dynamics of the guanidine-IV riboswitch RNA upon ligand binding. We employed single-molecule fluorescence resonance energy transfer (smFRET) to dissect the behaviors of the aptamer, terminator, and full-length riboswitch. Our findings indicated that the aptamer portion exhibited higher sensitivity to guanidine compared to the terminator and full-length constructs. Additionally, we utilized Position-specific Labelling of RNA (PLOR) combined with smFRET to observe, at the single-nucleotide and single-molecule level, the structural transitions experienced by the guanidine-IV riboswitch during transcription. Notably, we discovered that the influence of guanidine on the riboswitch RNA's conformations was significantly reduced after the transcription of 88 nucleotides. Furthermore, we proposed a folding model for the guanidine-IV riboswitch in the absence and presence of guanidine, thereby providing insights into its ligand-response mechanism.

*For correspondence:
liuyu_sjtu@sjtu.edu.cn

**Competing interest:** The authors declare that no competing interests exist.

## Introduction

Riboswitches, located at the 5'-untranslated region of mRNA, are capable of regulating gene expression through structural changes after binding to their specific ligands (*Jones and Ferré-D'Amaré, 2017*; *Mandal and Breaker, 2004*; *Serganov and Nudler, 2013*; *Sherlock and Breaker, 2020*). A typical riboswitch consists of an aptamer domain and an expression platform. The aptamer domain binds to a specific ligand, leading to structural changes and regulation of gene expression (*Garst et al., 2011*; *Kavita and Breaker, 2023*; *Winkler and Breaker, 2003*). To date, more than 55 classes of riboswitches have been identified that can sense a diverse range of ligands, including small metabolites, anions, cations, amino acids, and nucleotides (*Breaker, 2022*; *McCown et al., 2017*). Furthermore, riboswitches responding to the same ligands have been found to exhibit distinct consequences and structures. For instance, there are four types of guanidine riboswitches, namely guanidine-I,

guanidine-II, guanidine-III, and guanidine-IV riboswitch, which specifically bind to guanidine. Notably, the guanidine-IV riboswitch differs significantly from other guanidine riboswitches (*Battaglia and Ke, 2018*; *Huang et al., 2017a*; *Huang et al., 2017b*; *Lenkeit et al., 2020*; *Nelson et al., 2017*; *Salvail et al., 2020*). Specifically, the guanidine-IV riboswitch is able to enhance the transcription of *mepA* by forming an anti-terminator structure upon guanidine binding. This structure plays a crucial role in exporting guanidine and reducing its toxicity in cells, making the guanidine-IV riboswitch a potential target for antibiotics (*Kermani et al., 2018*). Although the atomic-resolution structures of the guanidine-IV riboswitch have not been solved, it is proposed that a kissing loop (KL) forms in the riboswitch in the presence of guanidine. This KL facilitates the dissociation of the terminator and promotes transcriptional read-through (*Lenkeit et al., 2020*; *Salvail et al., 2020*). KL or pseudoknot formation upon ligand binding is a common occurrence in various riboswitches, including adenine, guanine, ZTP, pre-Q1, fluoride, and c-di-AMP riboswitches (*Jones and Ferré-D'Amaré, 2017*; *Lenkeit et al., 2020*; *Salvail et al., 2020*). However, the mechanism by which KL affects gene expression in a transcriptional riboswitch, particularly during transcription, remains elusive.

The folding of RNA during transcription plays a critical role in its function, especially in riboswitches involved in gene regulation. Previous studies have investigated the co-transcriptional folding of various riboswitches using different techniques. For instance, *Frieda and Block, 2012* employed the optical-trapping assay to examine the co-transcriptional folding of an adenine riboswitch at the single-molecule resolution. *Helmling et al., 2017* and *Binas et al., 2020*, on the other hand, used NMR to map the transcriptional intermediates of the 2'-deoxyguanosine (2'-dG) and ZMP riboswitches at the single-nucleotide resolution. *Landgraf et al., 2022* utilized NMR and computational modeling to identify the ligand-sensing transcriptional window for the c-di-GMP and c-GAMP riboswitches. Hua et al. investigated the co-transcriptional folding of the twister ribozyme and the ZTP riboswitch by mimicking the folding process using helicases to dissociate an RNA-DNA hybrid (*Hua et al., 2020*; *Hua et al., 2018*). *Lou and Woodson, 2024*, *Uhm et al., 2018*, *Widom et al., 2018*, and *Yadav et al., 2022* employed the single-molecule fluorescence resonance energy transfer (smFRET) strategy to study the co-transcriptional folding of the *glmS* ribozyme, TPP, pre-Q1, and fluoride riboswitches by assembling transcriptional complexes. *Xue et al., 2023* monitored the folding of a growing SAM-VI riboswitch at the single-molecule resolution by FRET. Despite these previous investigations, there is lack of information regarding the conformational changes and ligand-sensing behavior of the guanidine-IV riboswitch during transcription. In this study, we used position-specific labeling of RNA (PLOR) to introduce a FRET pair, Cy3 and Cy5, at sites 78 and 35, respectively (*Liu et al., 2018*; *Liu et al., 2015*). Subsequently, we examined the ligand and $Mg^{2+}$ responses of the aptamer domain, terminator/anti-terminator, and full-length guanidine-IV riboswitch using smFRET. Our findings revealed that both guanidine and $Mg^{2+}$ facilitated the formation of a KL in the aptamer domain and significantly increased the KL-formed conformation in that domain. Interestingly, unlike the aptamer domain, the addition of guanidine and $Mg^{2+}$ only had a minimal effect on the formation of the KL-formed anti-terminator structure in the terminator/anti-terminator or full-length RNA. Additionally, we mimicked the conformational changes and ligand-sensing behavior from the aptamer domain to the terminator sequence of the guanidine-IV riboswitch during transcription using smFRET and PLOR. The formation of the KL increased as the riboswitch was transcribed up to 88 nucleotides (corresponding to the aptamer), but decreased as the riboswitch was transcribed from 88 to 105 nucleotides (corresponding to the terminator). Based on our data, it became evident that the transcriptional fate did not determined by the terminator sequence, but rather by the aptamer region, which exhibited distinct ligand-sensing abilities.

## Results

### The construction and characteristics of the guanidine-IV riboswitch

The *Clostridium botulinum* guanidine-IV riboswitch was investigated in this study. The secondary structures of the guanidine-IV riboswitch are depicted in *Figure 1A and B*. In the absence of the guanidinium cation ($Gua^+$), the terminator consists of the aptamer domain (in black) and poly-U tail (in red). This terminator is proposed to terminate transcription by dissociating RNA polymerase from the transcription system (*Figure 1A*). However, in the presence of $Gua^+$, the formation of a KL (in green, *Figure 1B*) hinders the formation of the complete terminator stem. This leads to the formation of the

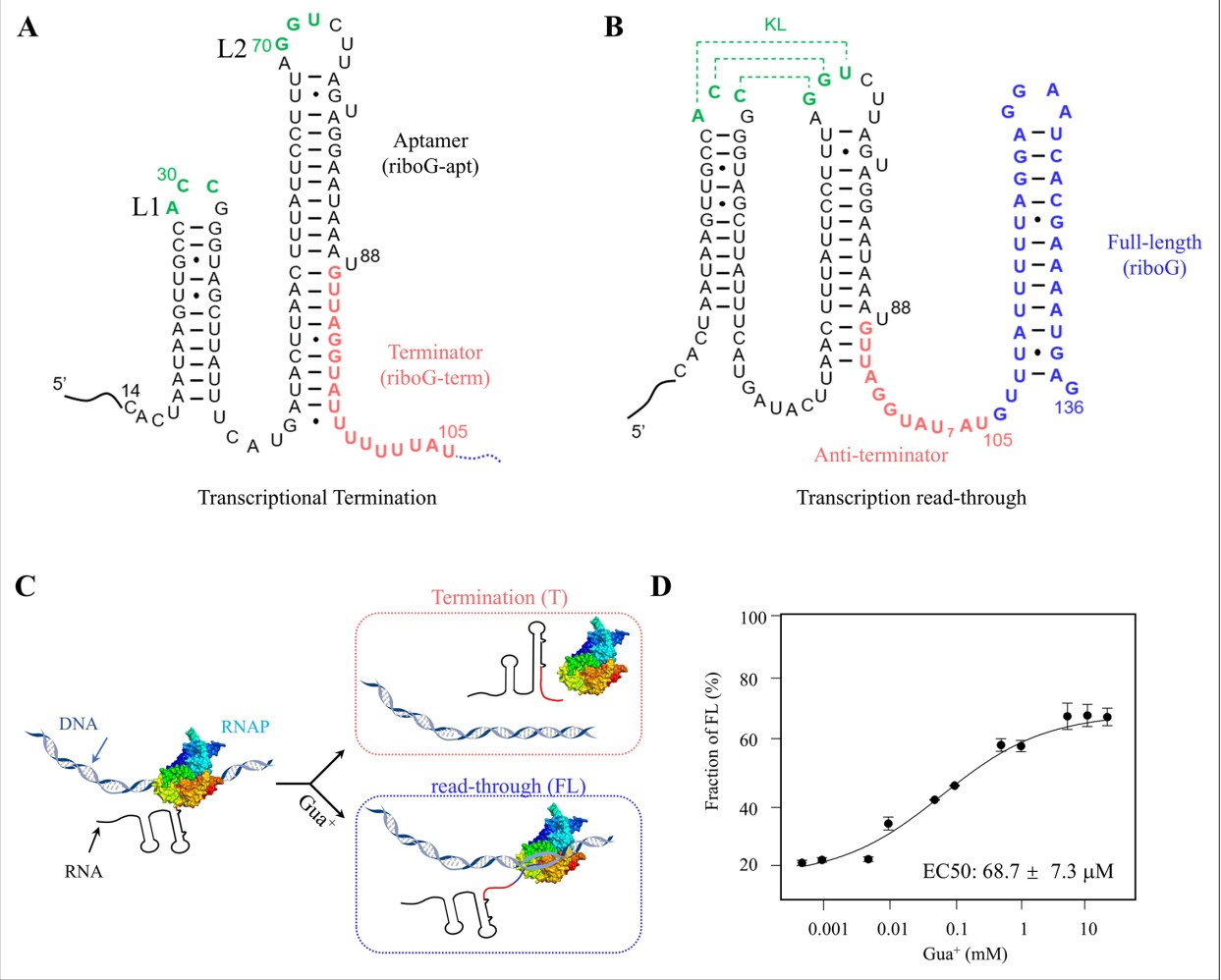

**Figure 1.** Characteristics of the *C. botulinum* guanidine-IV riboswitch. (**A, B**) The secondary structures of the transcriptional termination and read-through states of the guanidine-IV riboswitch. The full-length of the guanidine-IV riboswitch contains the aptamer domain (black), terminator (red) and the extended sequence (blue). The green nucleotides are involved in forming a KL in the anti-terminator state. (**C**) A transcriptional model of the guanidine-IV riboswitch. The DNA is shown in blue ribbons. The colors in the RNA are encoded as in (**A**) and (**B**). And the addition of Gua+ facilitates transcription read-through. (**D**) The percentages of the transcription read-through plotted with 0.5 μM–20.0 mM Gua+ at 6.0 mM Mg$^{2+}$ in the three independent transcription termination reactions. Data represent average ± SD from three replicates (n=3).

The online version of this article includes the following source data for figure 1:

**Source data 1.** Data for the graphs shown in *Figure 1D*.

anti-terminator conformation, and transcription of the full-length riboswitch comprises the extended nucleotides (in blue), in addition to the terminator/anti-terminator region (in red, *Figure 1B and C*). Consequently, the anti-terminator conformation activates the expression of the downstream genes (*Figure 1C and D*). In our study, we noted that the transcriptional read-through of the guanidine-IV riboswitch during the single-round PLOR reaction was sensitive to Gua+, exhibiting an apparent EC50 value of 68.7±7.3 μM (*Figure 1D*; *Chien et al., 2023*). This is comparable to the reported $K_D$ of approximately 64 μM that was determined by in-line probing (*Salvail et al., 2020*). Higher read-through efficiencies were detected at higher Gua+ concentrations, consistent with previous findings (*Lenkeit et al., 2020*). In order to probe the structural changes of different domains in the guanidine-IV riboswitch under various conditions, we applied PLOR to introduce Cy3 and Cy5 at sites 78 and 35 in the aptamer (riboG-apt), terminator/anti-terminator (riboG-term), and full-length guanidine-IV riboswitch (riboG). Sites 78 and 35 are close to the KL, making them suitable for monitoring the KL formation at different Mg$^{2+}$ and Gua+ concentrations.

## smFRET revealed the significant effects of $Mg^{2+}$ and $Gua^+$ on riboG-apt

smFRET is a powerful method that can detect multiple conformations by measuring the energy transfer efficiency ($E_{FRET}$) of a pair of fluorophores at the single molecule level. This technique has been extensively employed to investigate the conformational dynamics of RNA at the single-molecule level (*Manz et al., 2017*; *Uhm et al., 2018*). When the riboG-apt labeled with dual fluorophore-labeled (*Figure 2A*, *Figure 2—figure supplement 1*) was examined, a single peak ($E_{FRET}$ ~0.2) was observed in the presence of 0 mM $Mg^{2+}$ and 0 mM $Gua^+$ (*Figure 2B*). This peak is believed to correspond to the unfolded structure of riboG-apt, where the KL is not formed and thus, the labeled sites are distant from each other. Furthermore, in the presence of 2.0 mM $Mg^{2+}$, three peaks ($E_{FRET}$ ~0.2, 0.5, and 0.8) were observed, as revealed by hidden Markov modeling (*Figure 2C*). This suggests the coexistence of at least three structures when 2.0 mM $Mg^{2+}$ is present, with the peak at an $E_{FRET}$ of approximately 0.8 likely representing the KL formation. The peak characterized by an $E_{FRET}$ ~0.5 emerged at varying collection rates and was indicative of a more compact conformation compared to the unfolded structure, which we referred to as the pre-folded structure (*Figure 2D*, *Figure 2—figure supplements 2 and 3*). Notably, at 2.0 mM $Mg^{2+}$, these three states exhibited high dynamics, with transitions occurring particularly between the pre-folded and the unfolded states (*Figure 2B,C*, *Figure 2—figure supplements 2 and 3*). Our results demonstrate that the presence of $Mg^{2+}$ induces a conformational change in the unfolded riboG-apt, leading to the adoption of more compact pre-folded and folded conformations. The proportion of the folded conformation significantly increased to approximately 90% at 20.0 mM $Mg^{2+}$ (*Figure 2E*, *Figure 2—figure supplement 4G*). At high concentrations of $Mg^{2+}$, riboG-apt remained flexible and dynamic in its transition between states (*Figure 2—figure supplement 4*). However, at a higher concentration of 50.0 mM $Mg^{2+}$, the proportion of the pre-folded and unfolded conformations were more prevalent at 50.0 mM $Mg^{2+}$ than at 20.0 mM $Mg^{2+}$. This suggests that an excess of $Mg^{2+}$ may promote the pre-folded and even unfolded conformations.

$Gua^+$ also had the ability to promote the folded conformation ($E_{FRET}$ ~0.8) of riboG-apt. However, $Gua^+$ exhibited lower efficacy compared to $Mg^{2+}$, as evidenced by the proportion of the folded conformation being approximately 25% in the presence of 20.0 mM $Gua^+$. This value is significantly lower than the approximately 90% observed in the presence of 20.0 mM $Mg^{2+}$ (*Figure 2E*, *Figure 2—figure supplements 4G and 5F*). Our observations also revealed that riboG-apt exhibited pronounced dynamics, frequently transiting among the three states at 10.0–100.0 mM $Gua^+$ (*Figure 2—figure supplement 5E–H*). The presence of $Mg^{2+}$ was critical in facilitating the induction of the folded state by $Gua^+$. Notably, riboG-apt showed increased susceptibility to $Gua^+$ in the presence of $Mg^{2+}$ (*Figure 2F*, *Figure 2—figure supplement 6*). Upon the addition of 0.5 mM $Gua^+$ and 0.5 mM $Mg^{2+}$, the KL-folded conformation predominated in riboG-apt. The proportion of the KL-folded conformation increased to over 40%, which was comparable to the effect observed with a 20-fold increase in 0.5 mM $Gua^+$ in the absence of $Mg^{2+}$ (*Figure 2—figure supplements 5B and 6E*). Interestingly, at 0.5 mM $Mg^{2+}$, $Gua^+$ bound to riboG-apt with a $K_d$ of 286.0±18.1 μM (*Figure 2—figure supplement 6H*), higher than the $K_d$ of 64 μM that was determined by in-line probing in the presence of 20 mM $Mg^{2+}$ (*Salvail et al., 2020*). This difference is likely due to the low concentration of $Mg^{2+}$, which has a weak ability to help guanidine induced the KL conformation. In addition, we observed that riboG-apt became more stable as the concentration of $Gua^+$ increased, particularly in the presence of high $Mg^{2+}$ (*Figure 2—figure supplement 7*). Surprisingly, the proportion of the folded conformation hardly increased when $Mg^{2+}$ exceeded 2.0 mM in the presence of 1.0 mM $Gua^+$. Taking into account that $Gua^+$ was present in a positively charged state in the buffer employed for our FRET studies, we conducted additional experiments to investigate the impact of commonly used cations, including $Na^+$, $K^+$, and urea (a structural analog of guanidine). Interestingly, we observed minimal changes in the folded conformation of riboG-apt upon the addition of $Na^+$, $K^+$, or urea. These findings suggest that the effects of $Gua^+$ on the folded conformation are distinct from those of the tested cations. Furthermore, our results indicate that $Gua^+$ specifically binds to riboG-apt and induces structural changes as a ligand, rather than simply acting as a general cation or chaotrope (*Figure 2G*).

The highly conserved nucleotides surrounding the KL are crucial for its formation (*Lenkeit et al., 2020*). To test our hypothesis that the state with $E_{FRET}$ ~0.8 corresponds to the conformation with the KL, we preformed smFRET analysis on several mutations at these crucial nucleotides (*Figure 2—figure supplements 8–10*). Consistent with our expectations, the peaks with $E_{FRET}$ ~0.8 was significantly diminished in the riboG-G71C mutant, which features a single nucleotide mutation at site 71

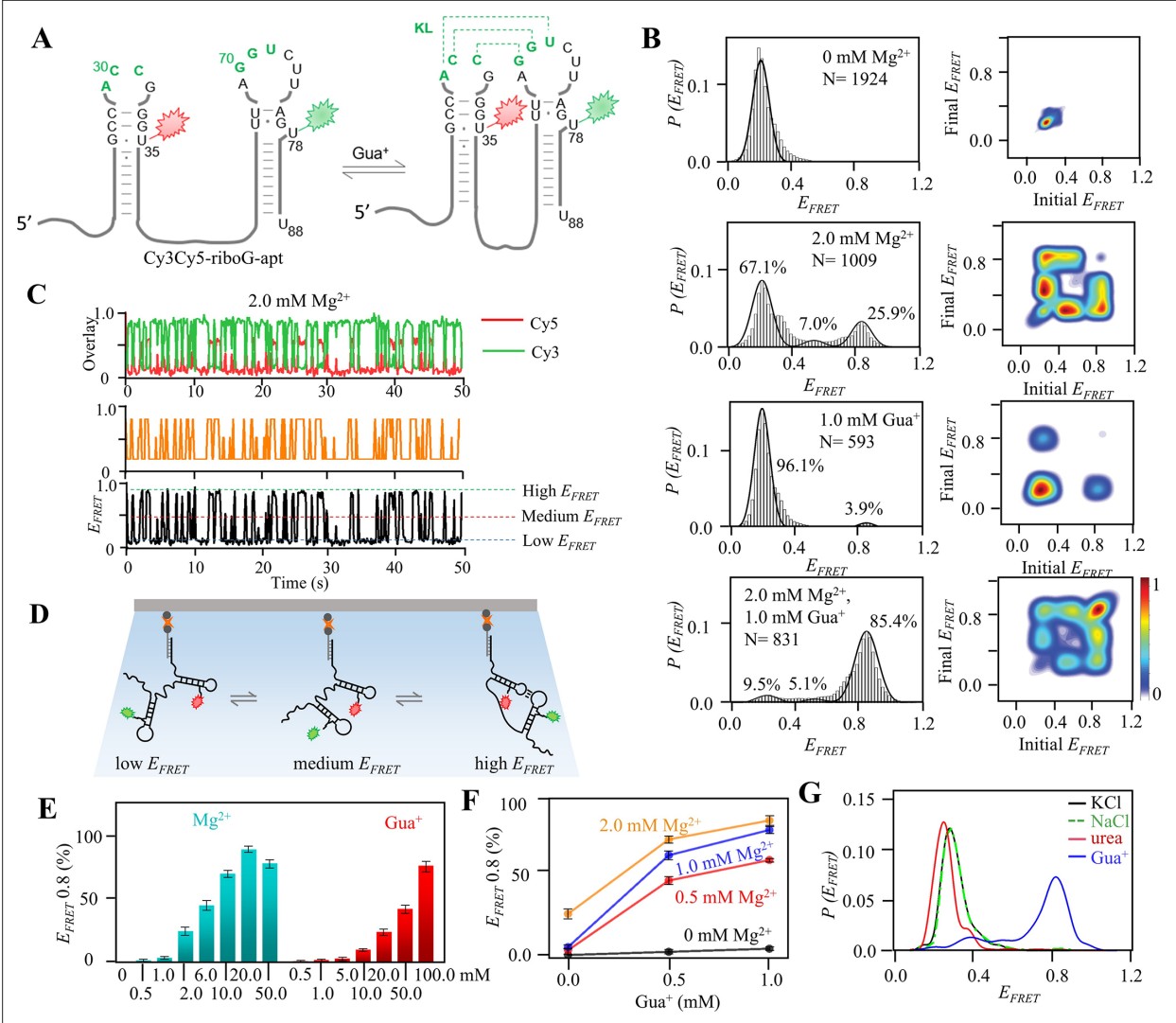

**Figure 2.** smFRET studies of post-transcriptional Cy3Cy5-riboG-apt at different concentrations of Mg²⁺ and Gua⁺. (**A**) The secondary structures of Cy3Cy5-riboG-apt at the unfolded (left) and the folded state (right). Cy3 and Cy5 are shown by green and red sparkles, respectively. (**B**) smFRET histograms and transition density plots for Cy3Cy5-riboG-apt at 0 mM Mg²⁺, at 2.0 mM Mg²⁺, at 1.0 mM Gua⁺, and at 2.0 mM Mg²⁺ and 1.0 mM Gua⁺. (**C**) HMM analysis of smFRET time trace of Cy3Cy5-riboG-apt at 2.0 mM Mg²⁺, the time resolution is 0.1 s. (**D**) Schematic diagram for three conformations of Cy3Cy5-labeled RNA in smFRET experiments. The Cy3Cy5-labeled RNA is hybridized with a biotinylated DNA and immobilized on the slides. (**E**) The percentages of the folded conformation (E$_{FRET}$ ~0.8) of Cy3Cy5-riboG-apt at 0–50.0 mM Mg²⁺ (cyan columns) and 0.5–100.0 mM Gua⁺ (red columns). Data represent average ± SD from three replicates (n=3). (**F**) The percentages of the folded conformation of Cy3Cy5-riboG-apt change with Gua⁺ at 0, 0.5, 1.0, and 2.0 mM Mg²⁺. (**G**) FRET histograms of Cy3Cy5-riboG-apt at 100.0 mM KCl (black curve), NaCl (green curve), urea (red curve), and guanidine (blue curve).

The online version of this article includes the following source data and figure supplement(s) for figure 2:

**Source data 1.** Data for the graphs shown in *Figure 2*.

**Figure supplement 1.** The schematic procedure of preparing Cy3Cy5-riboG-apt by 12 step-PLOR reaction for smFRET study.

**Figure supplement 2.** smFRET measurements of Cy3Cy5-riboG-apt at 2.0 mM Mg²⁺.

**Figure supplement 3.** smFRET measurements of Cy3Cy5-riboG-apt at different time resolution.

**Figure supplement 4.** smFRET measurements of Cy3Cy5-riboG-apt at 0–50.0 mM Mg²⁺.

**Figure supplement 5.** smFRET measurements of Cy3Cy5-riboG-apt at 0.5–100.0 mM Gua⁺.

**Figure supplement 6.** smFRET measurements of Cy3Cy5-riboG-apt at 0.01–10.0 mM Gua⁺ in the presence of 0.5 mM Mg²⁺.

**Figure supplement 6—source data 1.** Data for the graphs shown in *Figure 2—figure supplement 6H*.

**Figure supplement 7.** smFRET measurements of Cy3Cy5-riboG-apt at different Gua⁺ and Mg²⁺.

*Figure 2 continued on next page*

*Figure 2 continued*

**Figure supplement 8.** smFRET measurements of Cy3Cy5-riboG-G71C-apt and Cy3Cy5-riboG-C30G-G71C-apt.

**Figure supplement 9.** smFRET measurements of Cy3Cy5-riboG-U72C-apt and Cy3Cy5-riboG-A29G-U72C-apt.

**Figure supplement 10.** The G77C mutation perturbs the folding and function of riboG-apt.

**Figure supplement 10—source data 1.** Raw gels for *Figure 2—figure supplement 10C*.

**Figure supplement 10—source data 2.** Gels with labeled lanes used in *Figure 2—figure supplement 10C*.

(with 97% nucleotide conservation) in the KL (*Figure 2—figure supplement 8A and B*). It is worth noting that the C30G and G71C mutant, which were initially expected to restore a base pair in the KL, did not successfully bring about the anticipated peak of $E_{FRET}$ ~0.8 (*Figure 2—figure supplement 8C and D*). On the other hand, the riboG-U72C mutant exhibited a lower proportion at the state with $E_{FRET}$ ~0.8 than riboG-apt. However, the A29G and U72C mutations restored a base pair in the KL, as well as the formation of the KL (*Figure 2—figure supplement 9*). Furthermore, our investigation revealed that the G77C mutant, involving a single nucleotide mutation at a highly conversed site, 77 (with 97% nucleotide conservation), also hindered the formation of the KL (*Figure 2—figure supplement 10*). This finding aligns with previous research (*Lenkeit et al., 2020*) and the predicted second structure of G77C mutation by Mfold (*Zuker, 2003*).

## smFRET analysis revealed the slight effects of Mg$^{2+}$ and Gua$^+$ on riboG-term and full-length riboG

Unlike riboG-apt, riboG-term labeled with dual fluorophores exhibited the coexistence of three conformations, characterized by $E_{FRET}$ values of approximately 0.2, 0.5, and 0.8 (*Figure 3A*, *Figure 3—figure supplements 1–5*). This coexistence was observed even in the absence of Mg$^{2+}$ and Gua$^+$ (*Figure 3B*, *Figure 3—figure supplement 2A and 2B*). This indicates that monovalent ions in the buffer can facilitate the formation of stable guanidine-IV riboswitch. The sensitivity of RiboG-term to Gua$^+$ was significantly lower compared to riboG-apt. This is evidenced by the fact that, in the presence of 2.0 mM Mg$^{2+}$, the addition of 1.0 mM Gua$^+$ only resulted in an increase of approximately 4% the folded conformation for riboG-term, while it was approximately 60% for riboG-apt (*Figures 2B and 3B*). With the increase of either Mg$^{2+}$ from 0 to 50.0 mM or Gua$^+$ from 0 to 100.0 mM, less than 12% of the molecules transited dynamically among the three states of riboG-term (*Figure 3—figure supplements 2 and 3*). This finding suggests that the unfolded structure of riboG-term without the KL ($E_{FRET}$ ~0.2) is stable, and high concentrations of Mg$^{2+}$ or Gua$^+$ only slightly affect the KL formation (*Figure 3C*). In the presence of 0.5 mM Mg$^{2+}$, the structural change of riboG-term in response to increasing Gua$^+$ was negligible (*Figure 3D*, *Figure 3—figure supplement 4*). Surprisingly, in the presence of 1.0 mM Gua$^+$, the conformation dynamics of riboG-term became more evident as Mg$^{2+}$ increased. Moreover, at high Mg$^{2+}$ concentrations, such as 10.0 mM, the abundance of the folded conformation ($E_{FRET}$ ~0.8) significantly elevated with increasing Gua$^+$ (*Figure 3E*, *Figure 3—figure supplement 5*). This suggests that Gua$^+$ enhances the flexibility of riboG-term, potentially triggering its structural changes and facilitating the regulatory function of riboG in the presence of Mg$^{2+}$. In contrast to riboG-term, both its G71C and C30G-G71C mutants displayed a reduced proportion of the state with $E_{FRET}$ ~0.8. Remarkably, the fractions of $E_{FRET}$ ~0.8 remained unaffected by the addition of 1.0 mM Gua$^+$ in these mutants. Distinct from riboG-term, no structural transitions between states were observed in the two mutants (*Figure 3—figure supplement 6*). Regarding the U72C mutant of riboG-term, the mutation at the site 72 had a reduced impact on the KL conformation in the presence of 1.0 mM Gua$^+$ and 2.0 mM Mg$^{2+}$. However, the increased proportion of $E_{FRET}$ ~0.8 in the A29G-U72C mutant of riboG-term suggests that these mutations can restore the base-pairing between sites 29 and 72, as well as facilitate the formation of the KL (*Figure 3—figure supplement 7*).

In the presence of Mg$^{2+}$ and/or Gua$^+$, the FRET histogram of full-length riboG (*Figure 4A*, *Figure 4—figure supplement 1*) displayed a greater number of peaks compared to riboG-apt and riboG-term (*Figure 4B*, *Figure 4—figure supplement 2*). Additionally, the smFRET traces revealed structural transitions among the states, as shown in *Figure 4—figure supplements 2–5*. The presence of more structures in the full-length riboswitch compared to the aptamer domain is consistent with findings in other RNA molecules, such as the 2'-dG riboswitch (*Helmling et al., 2017*). Notably, upon the addition of 1.0 mM Gua$^+$ in the presence of 2.0 mM Mg$^{2+}$, the proportion of the new peak

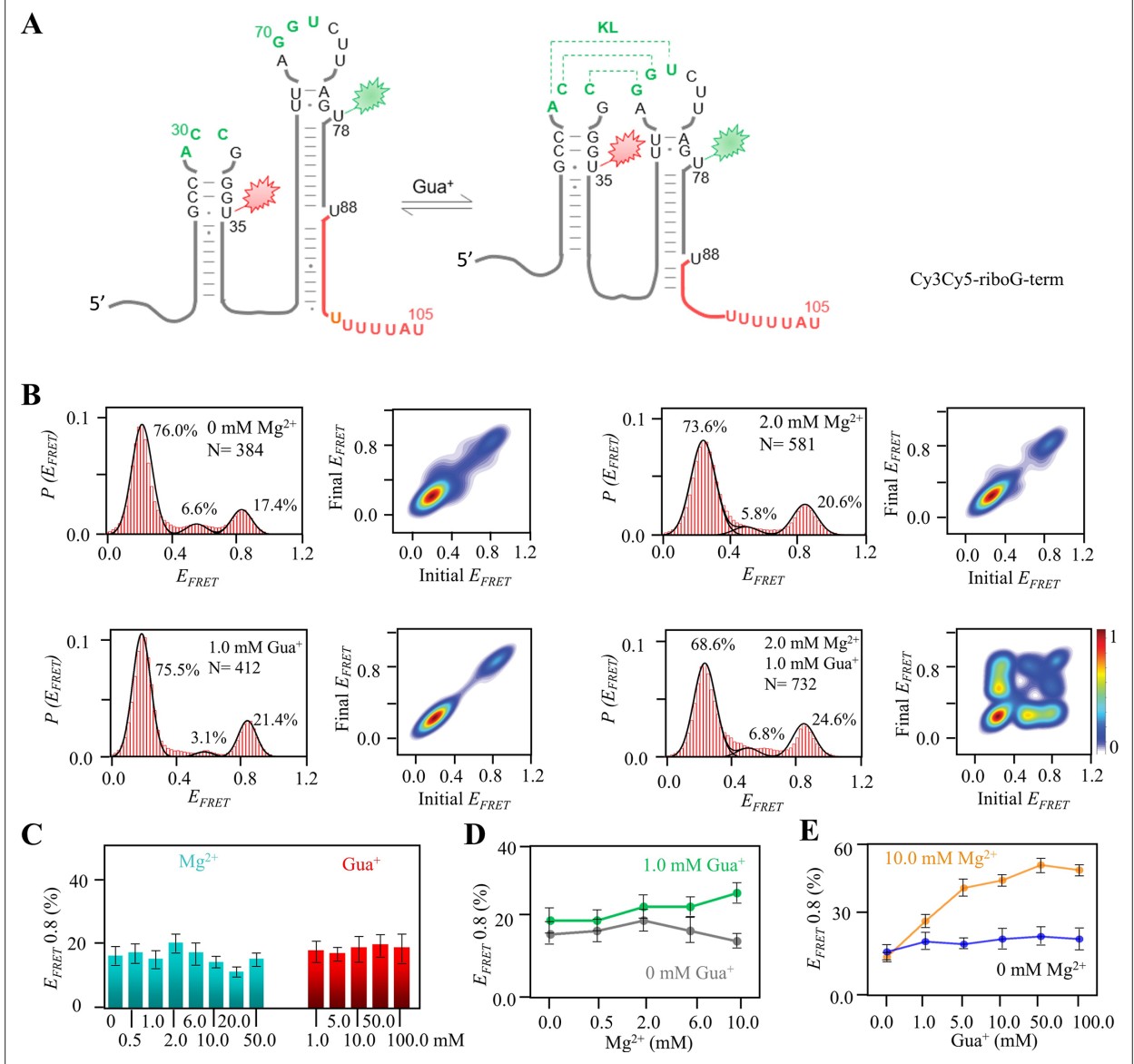

**Figure 3.** smFRET studies of post-transcriptional riboG-term at different concentrations of $Mg^{2+}$ and $Gua^+$. (**A**) The secondary structures of Cy3Cy5-riboG-term at the unfolded (left) and the folded state (right). (**B**) smFRET histograms and transition density plots of Cy3Cy5-riboG-term at 0 mM $Mg^{2+}$, at 2.0 mM $Mg^{2+}$, at 1.0 mM $Gua^+$, and at 2.0 mM $Mg^{2+}$ and 1.0 mM $Gua^+$. (**C**) The percentages of the folded conformation ($E_{FRET}$ ~0.8) of Cy3Cy5-riboG-term at 0–50.0 mM $Mg^{2+}$ (cyan columns) and 1–100.0 mM $Gua^+$ (red columns). (**D**) The percentages of the folded conformation of Cy3Cy5-riboG-term change with $Mg^{2+}$ at 0 (black) and 1.0 mM $Gua^+$ (green). (**E**) The percentages of the folded conformation of Cy3Cy5-riboG-term change with $Gua^+$ at 0 (blue) and 10.0 mM $Mg^{2+}$ (orange). Data represent average ± SD from three replicates (n=3).

The online version of this article includes the following source data and figure supplement(s) for figure 3:

**Source data 1.** Data for the graphs shown in *Figure 3*.

**Figure supplement 1.** The schematic procedure of preparing Cy3Cy5-riboG-term by 13 step-PLOR reaction for smFRET study.

**Figure supplement 2.** smFRET measurements of Cy3Cy5-riboG-term at 0–50.0 mM $Mg^{2+}$.

**Figure supplement 3.** smFRET measurements of Cy3Cy5-riboG-term at 1.0–100.0 mM $Gua^+$.

**Figure supplement 4.** smFRET measurements of Cy3Cy5-riboG-term at different $Gua^+$ and $Mg^{2+}$.

**Figure supplement 5.** smFRET measurements of Cy3Cy5-riboG-term at different $Gua^+$ and $Mg^{2+}$.

**Figure supplement 6.** smFRET measurements for Cy3Cy5-riboG-G71C-term and Cy3Cy5-riboG-C30G-G71C-term.

**Figure supplement 7.** smFRET measurements for Cy3Cy5-riboG-U72C-term and Cy3Cy5-riboG-A29G-U72C-term.

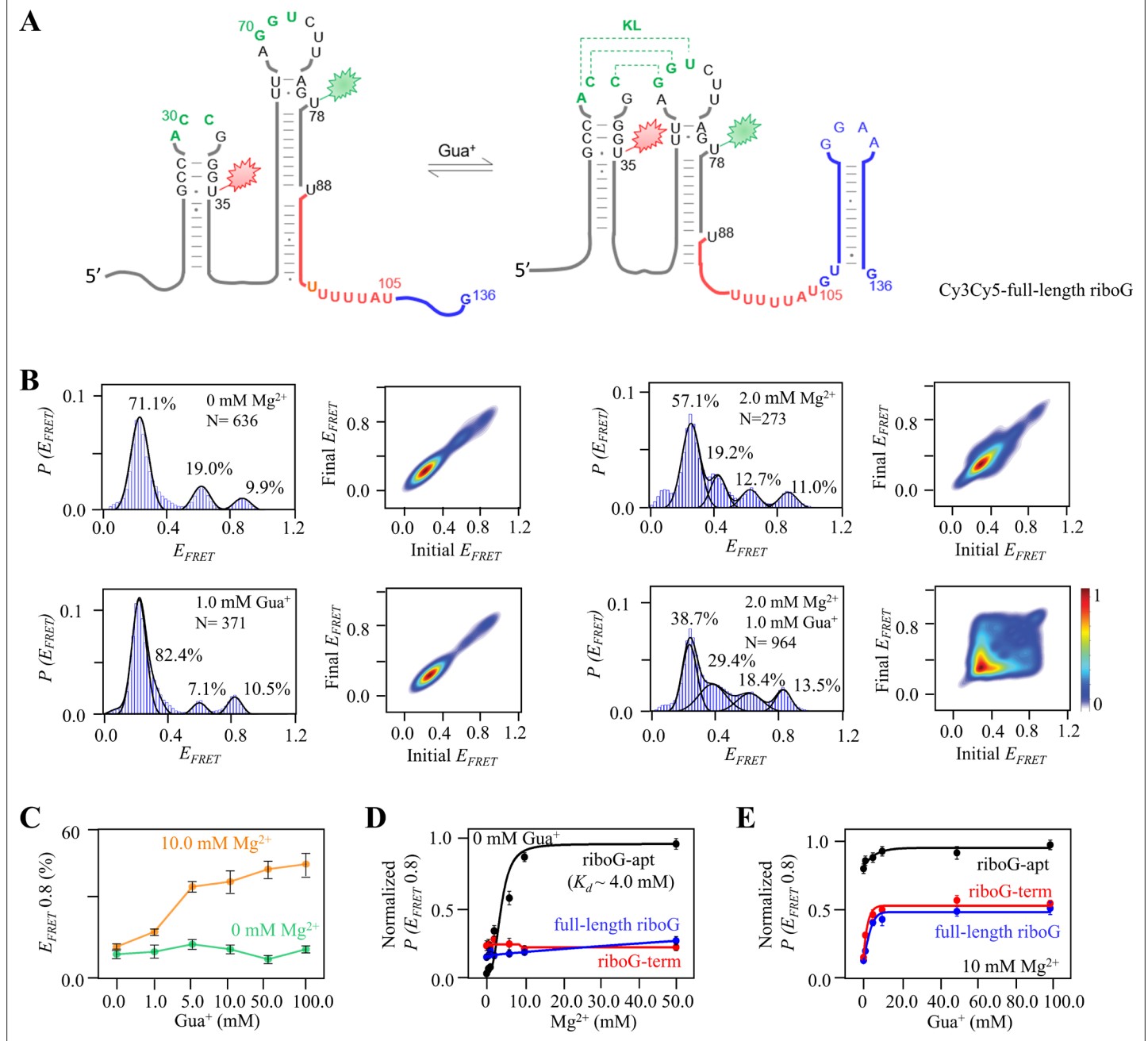

**Figure 4.** smFRET studies of the isolated full-length riboG at different concentrations of $Mg^{2+}$ and $Gua^+$. (**A**) The secondary structures of Cy3Cy5-full-length riboG at the unfolded (left) and the folded state (right). The full-length riboG contains the aptamer domain (black), terminator (red) and the extended sequence (blue). Cy3 and Cy5 are shown by green and red sparkles, respectively. (**B**) smFRET histograms and transition density plots for Cy3Cy5-full-length riboG at 0 mM $Mg^{2+}$, at 2.0 mM $Mg^{2+}$, at 1.0 mM $Gua^+$, and at 2.0 mM $Mg^{2+}$ and 1.0 mM $Gua^+$. (**C**) The percentages of the folded conformation of Cy3Cy5-full-length riboG change with $Gua^+$ at 0 (green) and 10.0 mM $Mg^{2+}$ (orange). (**D–E**) The normalized percentages of the folded conformation of Cy3Cy5-riboG-apt (black), Cy3Cy5-riboG-term (red) and Cy3Cy5-full-length riboG (blue) at 0–50 mM $Mg^{2+}$ (**D**) and 0–100.0 mM $Gua^+$ in the presence of 10.0 mM $Mg^{2+}$ (**E**). Data represent average ± SD from three replicates (n=3).

The online version of this article includes the following source data and figure supplement(s) for figure 4:

**Source data 1.** Data for the graphs shown in *Figure 4*.

**Figure supplement 1.** The schematic procedure of preparing Cy3Cy5-full-length riboG by 13 step-PLOR reaction for smFRET study.

**Figure supplement 2.** smFRET measurements of Cy3Cy5-full-length riboG at 0.5–50.0 mM $Mg^{2+}$.

**Figure supplement 3.** smFRET measurements of Cy3Cy5-full-length riboG at 1.0–100.0 mM $Gua^+$.

*Figure 4 continued on next page*

*Figure 4 continued*

**Figure supplement 4.** smFRET measurements of Cy3Cy5-full-length riboG at 1.0 mM Gua$^+$ and 2.0–10.0 mM Mg$^{2+}$.

**Figure supplement 5.** smFRET measurements of Cy3Cy5-full-length riboG at different Gua$^+$ and Mg$^{2+}$.

**Figure supplement 6.** smFRET measurements for Cy3Cy5-full-length riboG-G71C and Cy3Cy5-full-length riboG-C30G-G71C.

**Figure supplement 7.** smFRET measurements for Cy3Cy5-full-length riboG-U72C and Cy3Cy5-full-length riboG-A29G-U72C.

($E_{FRET}$ ~0.4) increased from 19% to 29%, and then decreased as the Mg$^{2+}$ concentrations increased from 2.0 to 10.0 mM (*Figure 4—figure supplements 2D and 4B–D*). In the absence of Gua$^+$, the addition of 0–50.0 mM Mg$^{2+}$ only slightly altered the proportion of the folded conformation ($E_{FRET}$ ~0.8) of the full-length riboG (*Figure 4—figure supplement 2*). Even in the presence of 1.0 mM Gua$^+$, an increase in Mg$^{2+}$ concentration from 0 to 10.0 mM had a slight effect on the proportion of the folded conformation (*Figure 4—figure supplement 4B–D*). However, the effect of Gua$^+$ on full-length riboG in the presence of 10.0 mM Mg$^{2+}$ was significantly more pronounced compared to its effect in the absence of Mg$^{2+}$ (*Figure 4C*, *Figure 4—figure supplement 5B-F*). The impact of Mg$^{2+}$ and Gua$^+$ on the folded conformation of full-length riboG showed similarities to their effects on riboG-term but had distinct differences compared to their effects on riboG-apt (*Figure 4D and E*). Overall, the effects of Mg$^{2+}$ and Gua$^+$ on the structures of full-length riboG and riboG-term were weaker compared to their effects on riboG-apt. Upon comparing the G71C and C30G-G71C mutants of the full-length riboG with their wild-type counterpart, it was observed that the wild-type adopted higher proportions of the state with $E_{FRET}$ ~0.8 (*Figure 4—figure supplement 6*). Regarding the U72C and A29G-U72C mutants of the full-length riboG, their behaviors with regards to the peak with $E_{FRET}$ ~0.8 were similar to that of their counterparts in riboG-term (*Figure 4—figure supplement 7*).

## Kinetic analysis of smFRET data to identify the folding pathway of the guanidine-IV riboswitch

The guanidine-IV riboswitch exhibits dynamic behavior, with its structural transitions in response to Mg$^{2+}$ and Gua$^+$ closely linked to transcription termination. The relative free energy ($\Delta\Delta G$) was calculated based on the state populations in the FRET histogram (*Figure 5A and B*). For riboG-apt, both the pre-folded and folded states displayed higher $\Delta\Delta G$ values compared to the unfolded state, indicating that the former states were less stable in the presence of 2.0 mM Mg$^{2+}$. However, the addition of 1.0 mM Gua$^+$ successfully reduced the $\Delta\Delta G$ of the folded state to a level lower than that of the unfolded state (*Figure 5A*). Meanwhile, only slight decreases in $\Delta\Delta G$ were observed for riboG-term and riboG upon the addition of 1.0 mM Gua$^+$ in the presence of 2.0 mM Mg$^{2+}$ (*Figure 5B*). The dwell time distributions were obtained from the single-molecule trajectories of FRET data for riboG-apt, riboG-term, and riboG (*Figure 5C–E*, *Figure 5—figure supplements 1–4*). The exponential decays of the dwell time were fitted to a first-order kinetic equation, allowing for the determination of dwell time and rate constants for the individual states of riboG-apt (*Figure 5C–E*). Based on these results, we proposed folding pathways to illustrate the transcriptional fate of the guanidine-IV riboswitch under different conditions (*Figure 5F*, *Figure 5—figure supplement 4*). The analysis of the dwell time data from single-molecule trajectories indicated that the folding process of riboG-apt was influenced by Mg$^{2+}$ and/or Gua$^+$. In the presence of 2.0 mM Mg$^{2+}$, there was inefficient structural switching from the unfolded riboG-apt ($\tau$~0.69 s) to the pre-folded state ($\tau$~0.11 s) or folded riboG-apt ($\tau$~0.40 s), resulting in the production of the terminator during transcription (red arrows, *Figure 5F*). Conversely, in the presence of 1.0 mM Gua$^+$, the primary switching of riboG-apt occurred between the unfolded ($\tau$~0.60 s) and folded states ($\tau$~1.04 s), which can subsequently be transcribed into the anti-terminator (blue arrows, *Figure 5F*).

## smFRET analysis revealed the Gua$^+$-sensitive transcription window of riboG

The sensitivity of Gua$^+$ varied among the post-transcriptional riboG-apt, riboG-term, and full-length riboG sequences. The folding and interactions of RNA during transcription play a critical role in its control functions for transcriptional riboswitch. To mimic how nascent riboG responds to Gua$^+$ and its regulatory effects during transcription, we utilized PLOR to control its transcription at a single-nucleotide pace. Simultaneously, we collected smFRET data from active transcription elongation

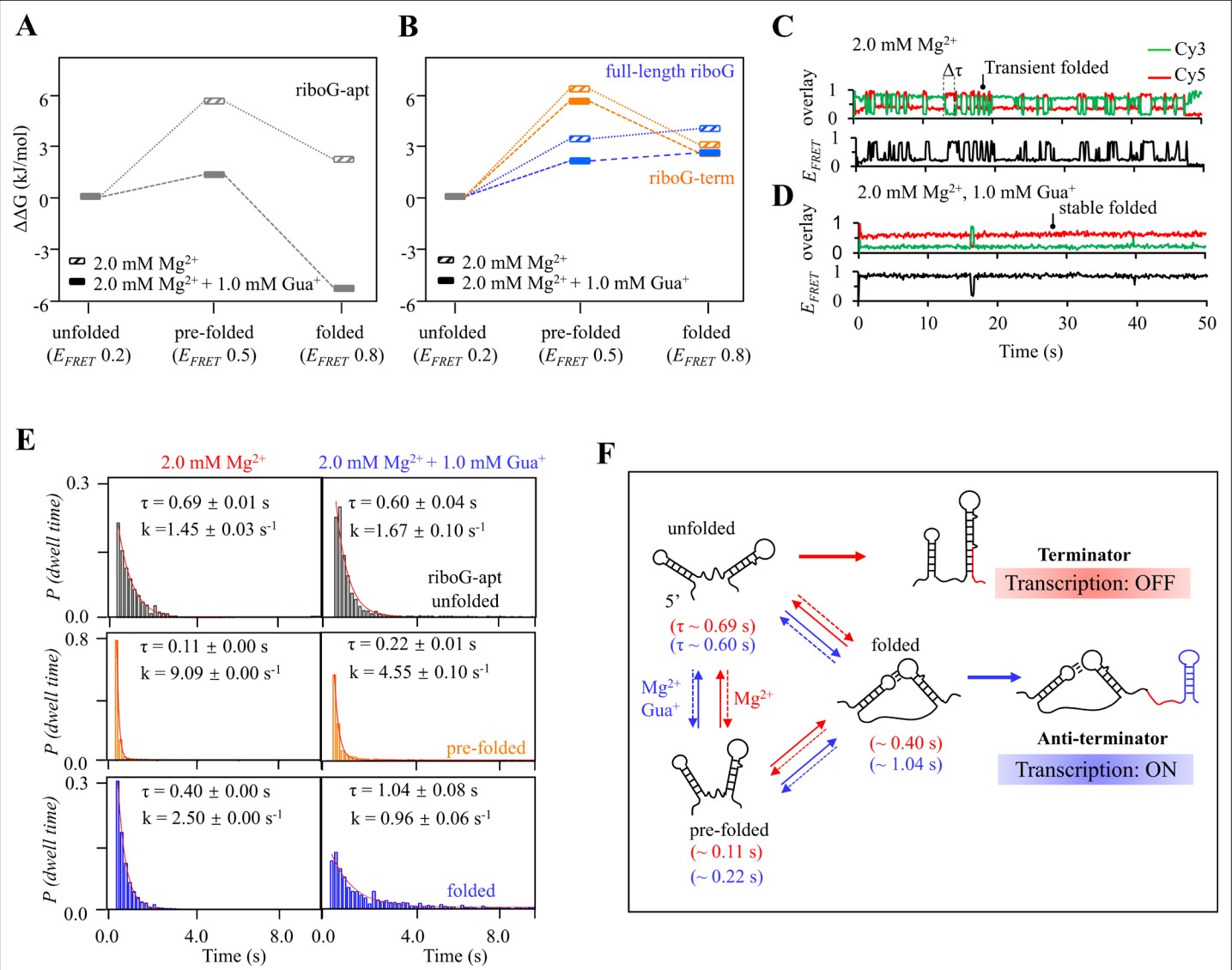

**Figure 5.** Relative free energy ($\Delta\Delta G$) and kinetics analysis of isolated riboG-apt, riboG-term and full-length riboG. (**A**) The relative free energy of the pre-folded and folded states of riboG-apt in the presence of 0 and 1.0 mM Gua+ at 2.0 mM Mg2+. The free energy of the unfolded state was referred as control. (**B**) The relative free energy of the pre-folded and folded states of riboG-term (orange) and full-length riboG (blue) in the presence of 0 and 1.0 mM Gua+ at 2.0 mM Mg2+. (**C–D**) Representative time traces of riboG-apt at 0 (**C**) and 1.0 mM Gua+ (**D**) in the presence of 2.0 mM Mg2+. $\Delta\tau$ is the dwell time. (**E**) The lifetime ($\tau$) and rate constant (k) of unfolded (black), pre-folded (orange) and folded (blue) states of riboG-apt was determined by exponential decays of the dwell time distributions. (**F**) Schematic diagram of riboG-apt folding and transcriptional processes at 2.0 mM Mg2+ in the presence of 0 mM Gua+ (red) and 1.0 mM Gua+ (blue).

The online version of this article includes the following figure supplement(s) for figure 5:

**Figure supplement 1.** Kinetics analysis of the unfolded, pre-folded and folded states of riboG-apt at 1.0 mM Mg2+.

**Figure supplement 2.** Kinetics analysis of isolated riboG-term and full-length riboG at 2.0 mM Mg2+.

**Figure supplement 3.** Kinetics analysis of isolated riboG-term and full-length riboG at 2.0 mM Mg2+ and 1.0 mM Gua+.

**Figure supplement 4.** Kinetics analysis of the unfolded, pre-folded or pre-folded 2 and folded states of riboG-apt, riboG-term and full-length riboG without and with guanidine at 2 mM Mg2+.

complexes (ECs) containing nascent RNA of different lengths (**Figure 6A,B**, **Figure 6—figure supplement 1**). PLOR is a liquid–solid hybrid phase transcription process that allows for pausing transcription at specific positions by omitting certain nucleotide triphosphates (NTPs). This paused transcription can then be restarted by adding the necessary NTPs. This unique 'pause–restart' transcriptional mode

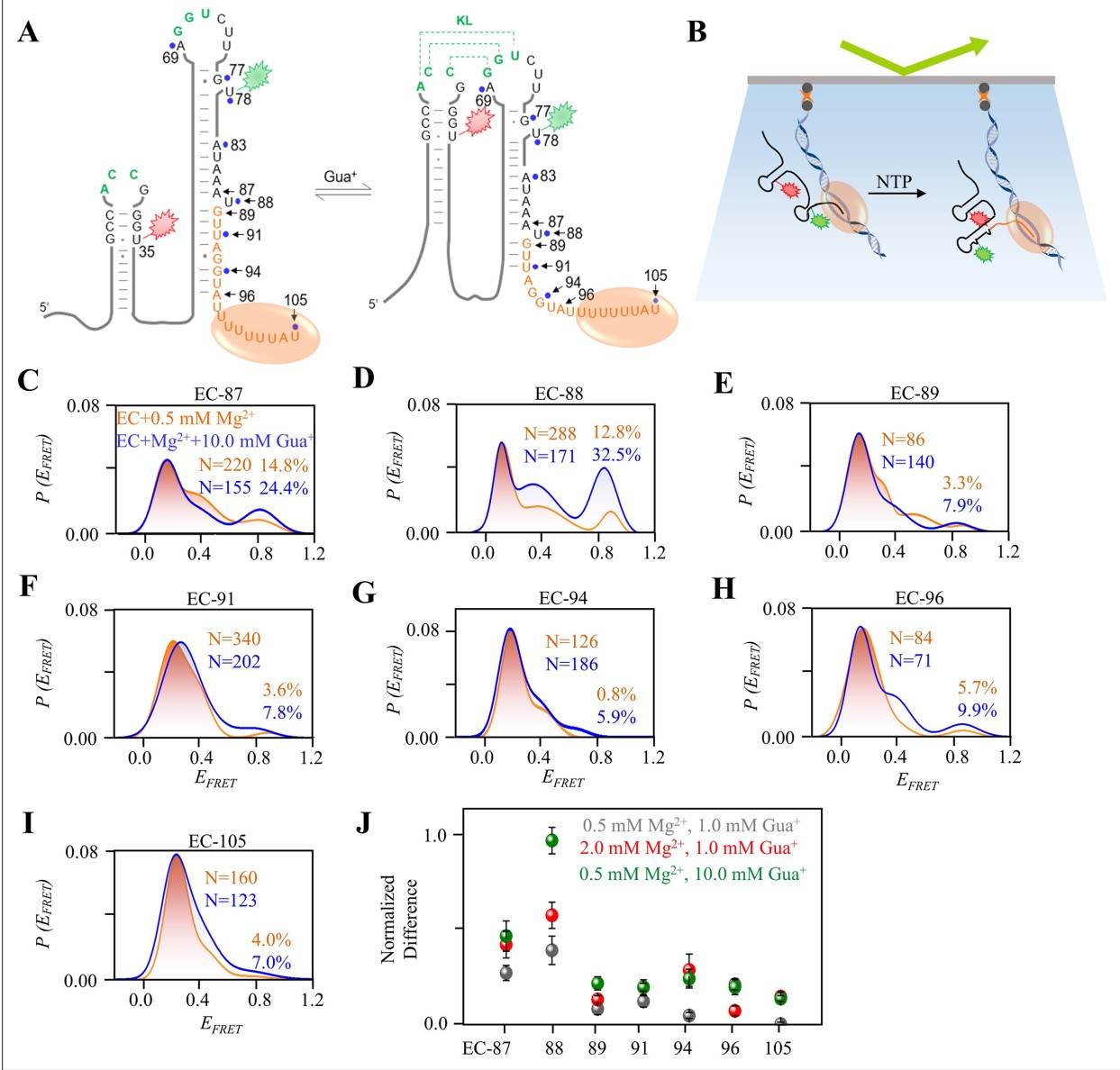

**Figure 6.** smFRET studies of nascent riboG in ECs without and with Gua$^+$ at 0.5 mM Mg$^{2+}$. (**A**) The ECs containing an RNA at the unfolded state (left) and the folded state (right). The pause sites of EC-87 to EC-105 are marked by black arrows, and the orange ellipse represents the T7 RNA polymerase. (**B**) Schematic diagram for smFRET experiment for ECs. The ECs were immobilized on the slides by their biotinylated DNA templates. (**C–I**) smFRET histograms for the EC-87 to EC-105 at 0 mM (orange) and 10.0 mM Gua$^+$ (blue) at 0.5 mM Mg$^{2+}$. (**J**) The normalized increase of the folded conformation (E$_{FRET}$ ~0.8) of ECs upon the addition of 1.0 mM Gua$^+$ at 0.5 mM Mg$^{2+}$ (grey), 10.0 mM Gua$^+$ at 0.5 mM Mg$^{2+}$ (green) and 1.0 mM Gua$^+$ at 2.0 mM Mg$^{2+}$ (red). Data represent average ± SD from three replicates (n=3).

The online version of this article includes the following source data and figure supplement(s) for figure 6:

**Source data 1.** Data for the graphs shown in *Figure 6J*.

**Figure supplement 1.** The schematic procedure of preparing EC-87 to EC-105 for smFRET study.

**Figure supplement 1—source data 1.** Raw gels for *Figure 6—figure supplement 1C*.

**Figure supplement 1—source data 2.** Gels for *Figure 6—figure supplement 1C*.

**Figure supplement 2.** smFRET studies of EC-87, EC-88, EC-89, and EC-91 at 0, 1.0 and 10.0 mM Gua$^+$ in the presence of 0.5 mM Mg$^{2+}$.

**Figure supplement 3.** smFRET studies of EC-94, EC-96 and EC-105 at 0, 1.0 and 10.0 mM Gua$^+$ in the presence of 0.5 mM Mg$^{2+}$.

**Figure supplement 4.** smFRET studies of EC-87, EC-88, EC-89, EC-91, EC-94, EC-96, and EC-105 at 0 and 1.0 mM Gua$^+$ in the presence of 2.0 mM Mg$^{2+}$.

*Figure 6 continued on next page*

*Figure 6 continued*

**Figure supplement 5.** Comparison of the folded-conformation percentages in EC-87, EC-88, EC-89, EC-91, EC-94, EC-96, and EC-105 at different Gua$^+$ in the presence of 0.5 mM Mg$^{2+}$ (**A**) and 2.0 mM Mg$^{2+}$ (**B**).

enables the evaluation of instant changes in the environment, such as the addition or removal of Gua$^+$ at specific steps, facilitating the monitoring of nascent RNA under varied conditions. To investigate the range of ligand-responsiveness in riboG, we employed PLOR to generate various transcriptional complexes with different RNA length, this allowed us to assess the impact of Gua$^+$ on the guanidine-IV riboswitch throughout the transcription process. We monitored the behavior of active transcription complexes containing guanidine-IV riboswitches with varying lengths of 87–105 nucleotides (nt) using smFRET (*Figure 6C–I*, *Figure 6—figure supplements 2–5*). To distinguish between the different complexes, we named them based on their nucleotide length, for example, the EC-87 for the 87 nt complex. We collected smFRET histograms for EC-87, EC-88, EC-89, EC-91, EC-94, EC-96, and EC-105 in the presence of 0.5 mM Mg$^{2+}$ and either 0 or 10.0 mM Gua$^+$ (*Figure 6C–I*, *Figure 6—figure supplements 2 and 3*). Three distinct peaks (E$_{FRET}$ ~0.2, 0.5, and 0.8) were observed, with the high-FRET peak suggesting the presence of a newly folded RNA structure, potentially containing a KL, as the transcript size exceeded 87 nt. In the presence of 0.5 mM Mg$^{2+}$, the proportion of the folded structure slightly decreased during transcription from 87 to 88 nt (from approximately 15% to 13%), but then sharply decreased to 3% at 89 nt. The addition of Gua$^+$ significantly increased the proportion of the folded state for EC-88 compared to EC-87. For longer ECs, such as EC-89, EC-91, and EC-94, the proportion of the folded state remained relatively stable (*Figure 6E–I*, *Figure 6—figure supplements 2 and 3*). At 0.5 and 2.0 mM Mg$^{2+}$, the proportions of the peaks changed differently, but EC-88 exhibited more dramatic changes than other ECs upon the addition of Gua$^+$ (*Figure 6J*, *Figure 6—figure supplements 2–5*). These results suggest that the optimal formation of KL occurs during transcription of the 88 nt nascent RNA, and a drastic structural switch takes place between 88 and 89 nt in the presence of Mg$^{2+}$. Interestingly, the addition of 1.0 mM Gua$^+$ induced more pronounced changes in the majority of ECs at higher Mg$^{2+}$ concentrations (*Figure 6J*, *Figure 6—figure supplements 4 and 5*). These findings suggest that the ECs exhibit a greater propensity for structural transitions upon the addition of Gua$^+$ at high Mg$^{2+}$ concentrations. Furthermore, under conditions of 0.5 mM Mg$^{2+}$ and 10.0 mM Gua$^+$, the proportions of folded structures (E$_{FRET}$ ~0.8) in EC-88 and EC-105 were approximately 36% and 14% lower, respectively, compared to those observed in the isolated post-transcriptional riboG-apt and riboG-term (*Figure 6D,I*, *Figure 2—figure supplement 6* and *Figure 3—figure supplement 4F*). The T7 RNA polymerase (RNAP) sequestered about 8 nt of the nascent RNA, preventing the EC-88 construct from forming the P2 stem (*Durniak et al., 2008*; *Huang and Sousa, 2000*; *Lubkowska et al., 2011*; *Tahirov et al., 2002*; *Wang et al., 2022*; *Yin and Steitz, 2002*). Consequently, a pseudoknot structure potentially formed instead of the expected KL. This distinction may account for the observed heterogeneity between EC-88 and riboG-apt.

## Gua$^+$ modulated the transcription of the guanidine-IV riboswitch at specific sequences

To investigate the regulatory functions of the nascent riboswitch at different lengths during transcription, we performed a transcriptional termination assay of the guanidine-IV riboswitch using PLOR (*Figure 7A*, *Figure 7—figure supplement 1*). Specifically, we carefully designed eight PLOR experiments to temporarily pause the transcription at nucleotides 69, 77, 78, 83, 88, 91, 94, and 105, respectively, before transcribing the remaining RNA fragment in the last step, following strategies 1–8 (*Figure 7B*, *Figure 7—figure supplement 1B and 1C*). In the absence of Gua$^+$, there was minimal change in transcriptional termination among strategies 1–7, but a sharp decrease of approximately 20% was observed for strategy 8 (*Figure 7C and D*). However, with the addition of Gua$^+$ in the last step, there was little variation in transcriptional read-through of the guanidine-IV riboswitch among strategies 1–5, but a dramatic decrease in anti-termination from strategy 6 to strategy 8 (*Figure 7C and D*). The detailed procedures of strategies 1–8 were shown in *Figure 7—figure supplement 1*. Furthermore, we observed a significant increase in read-through upon the addition of Gua$^+$ in our single-round transcription reactions when the RNA length reached 88 nt (*Figure 7—figure supplement 1D*). Additionally, a slight increase in read-through was observed at the pausing site 105, which is consistent with the smFRET data presented in *Figure 6J*. This indicates that Gua$^+$ can still bind and

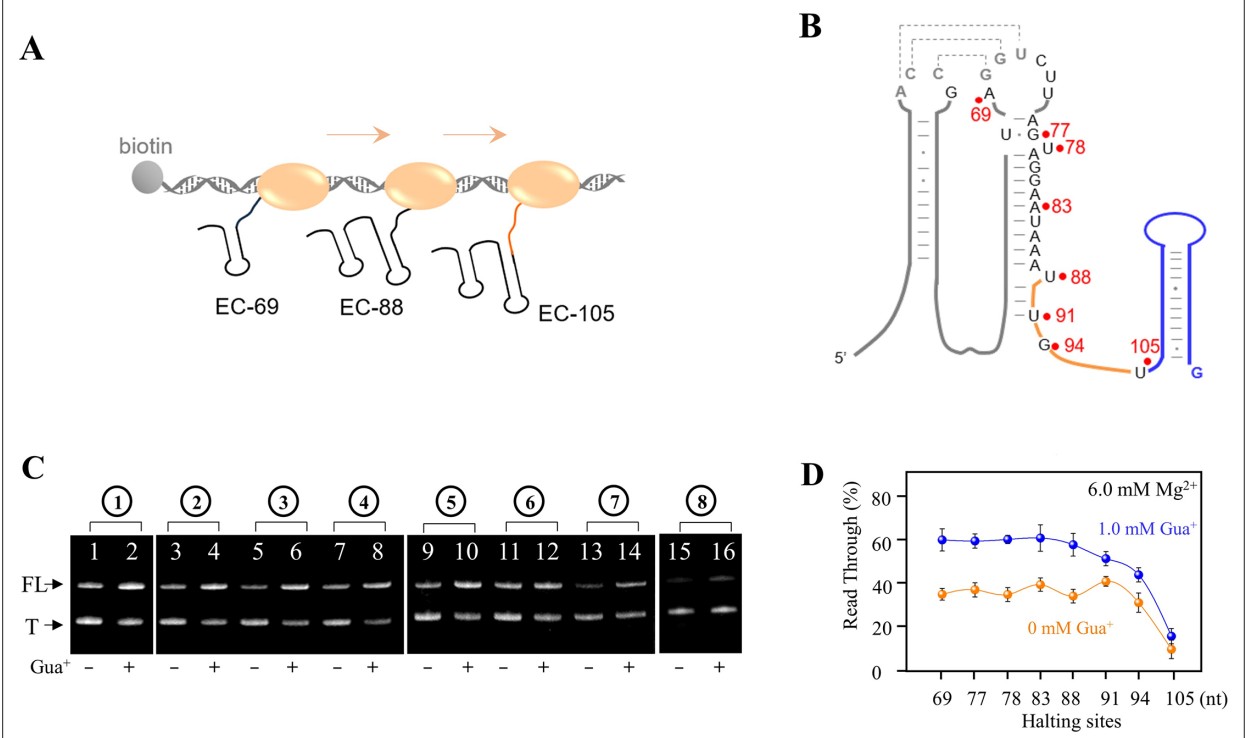

**Figure 7.** Ligand-mediated conformational switching of riboG during transcription. (**A**) The schematic diagram of synthesizing ECs with nascent RNAs by in vitro transcriptional pause using PLOR reactions. The orange ellipse represents T7 RNA polymerase. (**B**) The secondary structures of full-length riboG. The halting sites of EC-69, EC-77, EC-78, EC-83, EC-88, EC-91, EC-94, and EC-105 are marked by red dots. (**C**) PAGE images of the crude products collected at the last step of individual PLOR strategy in the absence and presence of 1.0 mM Gua⁺. FL and T are represented for the read-through and terminated RNA. The detail strategies were depicted in *Figure 7—figure supplement 1B*. (**D**) The transcriptional read-through fractions were plotted with the last halting sites in 9 step-, 11 step-, 12 step-, 13 step-, 14 step-, 15 step-, 16 step-, and 17 step-PLOR. The orange and blue curves are in the absence and presence of 1.0 mM Gua⁺. Taking into consideration that the 17 step-PLOR reaction exhibited a pause within the terminator region, resulting in a significant amount of terminated product at step 16, crude products from steps 16 and 17 were collected for (**C**) and (**D**) of the 17 step-PLOR reaction (Lanes 15 and 16 in C). Data represent average ± SD from three replicates (n=3).

The online version of this article includes the following source data and figure supplement(s) for figure 7:

**Source data 1.** Raw gels for *Figure 7C*.

**Source data 2.** Gels for *Figure 7C*.

**Source data 3.** Data for the graphs shown in *Figure 7D*.

**Figure supplement 1.** In vitro transcriptional termination assay of the guanidine-IV riboswitch by PLOR.

induce slight structural changes in the guanidine-IV riboswitch, even if its full terminator sequence has been transcribed. Our findings from in vitro transcriptional termination assays and smFRET results, indicating that EC-91 exhibits a more stable fold with a longer stem, making it less sensitive to environmental changes compared to EC-88 (*Figure 6F*, *Figure 7—figure supplement 1D*).

## Discussion

The guanidine-IV riboswitch exhibits similarities to the guanidine-I riboswitch in gene regulatory mechanism, functioning as a transcriptional riboswitch. Structurally, it resembles the guanidine-II riboswitch through the formation of loop-loop interactions upon binding to guanidine (*Battaglia and Ke, 2018*; *Huang et al., 2017a*; *Huang et al., 2017b*; *Lenkeit et al., 2020*; *Nelson et al., 2017*; *Reiss and Strobel, 2017*; *Salvail et al., 2020*). In this study, we have demonstrated that both Mg²⁺ and Gua⁺ play critical roles in promoting structural folding and transcriptional anti-termination in the guanidine-IV riboswitch. Based on our findings, we propose a model illustrating the termination/anti-termination process of the guanidine-IV riboswitch as transcription progresses (*Figure 8*). In the presence of Mg²⁺, the proportion of the anti-terminator induced by Gua⁺ increases as transcription progresses up to

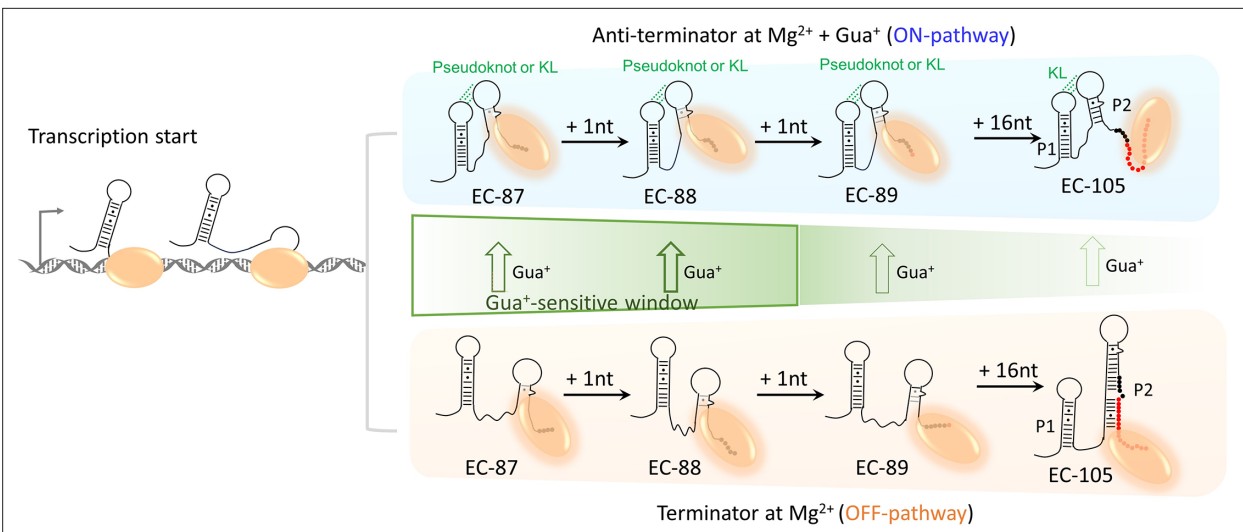

**Figure 8.** Folding model of riboG in the absence and presence of guanidine. The folding pathways of anti-terminator conformation in the presence of Gua$^+$ and terminator conformation in the absence of Gua$^+$ are highlighted in blue and orange, respectively. The H-bonds in the KL or pseudoknot are shown as dotted green lines. The nucleotides from EC-87 to EC-88 are shown as black dots, the nucleotides in termination sequence are shown as red dots in EC-89 and EC-105, and DNA templates from EC-87 to EC-105 are not displayed in the model. The T7 RNAP is shown as orange ellipse. The DNA templates are shown as gray ribbons. Green arrows represent the direction of structural switching in native riboG upon the addition of the Gua$^+$. The thicker the green arrow, the higher the switching percentages. And the Gua$^+$-sensitive transcription window of riboG from EC-87 to EC-88 is boxed by green lines.

EC-88. However, after EC-88, a significant decrease is observed up to EC-105. The elongation of transcription from EC-88 to EC-105 leads to impaired pseudoknot or KL formation and an increase in terminator abundance. This incremental switch from the anti-terminator to the terminator from EC-88 to EC-105 is influenced by the elongating P2 structure (*Figure 8*). The competition between the P2 and KL structures determines the proportions of the terminator and the anti-terminator in the guanidine-IV riboswitch. The stability of the terminator is enhanced by the presence of P2, and the increase in base pairs within P2 strengthens the terminator as transcription proceeds. On the other hand, the stability of the anti-terminator is dependent on the formation of pseudoknot or KL. The elongation of P2 shortens the flexible linker between P1 and P2, which creates steric hindrance for KL after EC-88. Furthermore, the addition of Gua$^+$ facilitates the functional switch from the terminator to the anti-terminator for constructs ranging from EC-87 to EC-105. Here, we propose a narrow Gua$^+$-sensitive transcription window for the guanidine-IV riboswitch, with a dramatic switch observed during the extension from EC-87 to EC-88 upon Gua$^+$ addition (shown in a green box, *Figure 8*). It is worth noting that the ligand-sensitive sequence for the guanidine-IV riboswitch is located within the aptamer domain, spanning 88 nucleotides.

Our findings demonstrate that the guanidine-IV riboswitch possesses a distinct binding window that can be modulated by guanidine. The mechanism of anti-termination supports a model in which the formation of a long-range KL prevents the assembly of a rigid terminator. This distinguishes the guanidine-IV riboswitch from typical transcriptional riboswitches that rely on overlapping sequences and base-pair rearrangements for functional change (*Strobel et al., 2019*; *Yadav et al., 2022*; *Zhao et al., 2017*). Moreover, our results reveal the existence of at least three distinct states in the guanidine-IV riboswitch, including a folded state characterized by the presence of a KL that plays a critical role in ligand binding and transcriptional regulation. Importantly, the formation of the long-range KL in the aptamer domain is specifically responsive to guanidine, as other cations such as K$^+$, Na$^+$, and urea do not produce the same effect. While guanidine contributes to structural switching in the aptamer domain, its impact on the terminator or full-length guanidine-IV riboswitch is comparatively weak. Additionally, our data indicate that guanidine induces less significant changes once the RNA polymerase has transcribed the aptamer domain. It is worth noting that the RNAP utilized in our study is T7 RNAP, which exhibits distinct characteristics compared to bacterial RNAP in terms of transcriptional speed, dynamics, and interactions. However, Xue et al. have reported similarities between

T7 and *E. coli* RNAP in the folding of nascent RNA. Additionally, Lou and Woodson have provided valuable insights into the co-transcriptional folding of the *glmS* ribozyme using T7 RNAP (*Xue et al., 2023*; *Lou and Woodson, 2024*).

In the transcriptional mimicking assay, the nascent RNA displayed a weaker response to guanidine compared to the post-transcriptional counterparts. The guanidine sensitivity of EC-87 indicates that guanidine regulation for the guanidine-IV riboswitch occurs before transcription of the aptamer domain is completed. Additionally, the sensitivity to guanidine gradually increases in the RNA within ECs until EC-88, after which it decreases. In other words, there exhibits a limited transcriptional window for the guanidine-IV riboswitch to dynamically respond to guanidine, and the inclination to form the terminator increases after transcribing 88 nucleotides, resulting in a decrease in the abundance of the KL-folded conformation. This aligns with previous findings that the transcription decision of a transcriptional riboswitch is made prior to terminator formation (*Sherwood and Henkin, 2016*; *Wickiser et al., 2005*). Nevertheless, the ligand-sensitive windows of riboswitches during transcription vary. In a study conducted by *Helmling et al., 2017* using NMR spectroscopy, they proposed a broad transcriptional window for deoxyguanosine-sensing riboswitches, whereby the ligand binding capability gradually diminishes over several nucleotide lengths. However, more recent research by *Binas et al., 2020* and *Landgraf et al., 2022* on riboswitches sensing ZMP, c-di-GMP, and c-GAMP revealed a narrow window with a sharp transition in binding capability, even with transcript lengths differing by only one or three nucleotides. In line with the findings for the c-GAMP-sensing riboswitch, our study on the guanidine-IV riboswitch also demonstrated a sharp transition in binding capability with just a single nucleotide extension. Our data also suggest that guanidine can induce slight structural changes in the guanidine-IV riboswitch even after the terminator domain has been transcribed and stabilized. Similar observations have been reported for other riboswitches, although the transcriptional window sensitive to the ligand differs among riboswitches (*Wickiser et al., 2005*).

## Materials and methods

### DNA template preparation

The biotin labeled DNA templates were prepared by PCR as described earlier (*Liu et al., 2018*; *Liu et al., 2015*). The sequences of the DNA templates are listed in *Supplementary file 1, table S1*. The primers used in PCR were ordered from Sangon Biotech (Shanghai, China). The DNA products generated from PCR were purified by 12% denaturing PAGE (polyacrylamide electrophoresis) before immobilized on streptavidin-coated agarose beads (Smart-Life Science, Changzhou, China) in the buffer (40 mM Tris-HCl, 100 mM $K_2SO_4$, 6 mM $MgSO_4$, pH 8.0) as described earlier (*Liu et al., 2018*). The solid-phase DNA templates were stored at 4 °C before usage.

### Cy3Cy5-labeled riboG-apt, riboG-term and full-length riboG synthesis

The Cy3Cy5-labeled RNA used in smFRET were synthesized by PLOR as described earlier (*Liu et al., 2018*; *Liu et al., 2015*), and RNA sequences were depicted in *Supplementary file 1, tables S1, S2*. The schemes and reagents of synthesis of Cy3Cy5-riboG-apt, Cy3Cy5-riboG-term and Cy3Cy5-full-length riboG are listed in *Supplementary file 1,tables S3, S4 and S5*. For Cy3Cy5-riboG-apt synthesis, 10 µM DNA-beads gently shook with 10 µM T7 RNAP, 0.96 mM ATP, 0.96 mM GTP and 96 µM UTP in the buffer (40 mM Tris-HCl, 100 mM $K_2SO_4$, 6 mM $MgSO_4$, 10 mM DTT, pH 8.0) at 37 °C for 15 min in the first step. The reaction mixture was filtered by solid-phase extraction and rinsed for 3~4 times in the buffer (40 mM Tris-HCl, 6 mM $MgSO_4$, pH 8.0). And the bead-rinsing procedure was performed before the addition of NTPs in each step except noted. The residual steps were proceeded in the elongation buffer (40 mM Tris-HCl, 6 mM $MgSO_4$, 10 mM DTT, pH 8.0) at 25 °C for 10 min with the addition of different NTP mixture at individual step as listed in *Supplementary file 1, table S3*. The Cy3Cy5-labeled riboG-apt was collected at the last step, purified by 12% denaturing PAGE and reversed phase HPLC loaded with C8 column (4.6*250 mm, Cat. No. EXL-122–2546 U, Phenomenex Luna, USA) as described earlier. The Cy3Cy5-riboG-term and Cy3Cy5-full-length riboG were synthesized as Cy3Cy5-riboG-apt except different NTPs were added. The purified RNA was stored at –80 °C before usage. T7 RNAP was utilized in the PLOR and in vitro transcription reactions except noted.

## Preparation of transcription complexes containing RNA at different lengths

The RNA sequences in the transcription complexes (ECs) are listed in *Supplementary file 1, table S6*, and the ECs were collected at different steps of PLOR as listed in *Supplementary file 1,table S7*. For example, EC-87 was obtained as the following procedures: took 10 μL 10 μM DNA-beads from 200 μL of the PLOR reaction at step 14 and incubated with 1 μL 1 mM biotin at 25 °C for 20 min. The addition of biotin was to displace the biotin-labeled DNA template on the streptavidin-labeled beads, filtered by solid-phase exchange and collected the elution containing EC-87. EC-88, 89, 91, 94, 96, and 105 were obtained at steps15, 16, 17, 18, 20, and 21 similarly as EC-87 (*Supplementary file 1, table S7*).

## smFRET experiments and data analysis

0.5 μM Cy3Cy5-riboG-apt, Cy3Cy5-riboG-term or Cy3Cy5-full-length riboG were hybridized with 0.75 μM biotinylated 13 nt-DNA by incubating at 90 °C for 5 min and then cooled to room temperature in the T50 buffer (10 mM Tris-HCl, 50 mM NaCl, pH 8.0). The sequence of the biotinylated DNA is listed in *Supplementary file 1, table S1*. The hybridized Cy3Cy5-labeled RNA was then diluted to 10~100 pM for the subsequent smFRET measurements. Polymer-coated quartz slides were immersed at the 0.1 mg/μL biotin-PEG and 0.3 mg/μL m-PEG for 2 h (*Roy et al., 2008*), then coated with 0.2 mg/mL streptavidin for 5 min. 10~100 pM Cy3Cy5-labeled RNA or ECs in the buffer (10 mM Tris-HCl, 50 mM NaCl, 0~100 mM guanidine, 0~100 mM $Mg^{2+}$) was injected and then rinsed 3 times to remove the free RNA or ECs. 3 mM Trolox (MCE, USA), 5 mM 3,4-protocatechuic acid (PCA, Shanghai Aladdin Bio-Chem Technology Co., China) and 5 nM protocatechuate dioxygenase (PCD) was added to reduce the photobleaching (*Aitken et al., 2008*). smFRET experiments were performed by using an objective-type total internal reflection fluorescence (TRIF) microscopy and an inverted microscope (Eclipse Ti, Nikon, Japan) at 20 °C, with FRET marker dyes Cy3 (donor) and Cy5 (acceptor) was excited by 532 nm and 640 nm laser, respectively. Open-sourced software iSMS (*Preus et al., 2015*) was used to process the single-molecule videos to extract the time-dependent signals. Time resolution of 100ms was used for each single-molecule video collection excepted noted. Single-molecule trajectories were identified by Deep FRET (*Thomsen et al., 2020*). The FRET efficiency, $E_{FRET}$ was calculated using the equation:

$$E_{FRET} = I_A/(I_A + I_D).$$

where $I_A$ and $I_D$ are acceptor and donor fluorescence intensity, respectively. We used empirical Bayesian method of the hidden Markov modelling (HMM) package vbFRET to estimate the FRET states and the time points of transitions (*Bronson et al., 2009*). The transition density plots (TDP) were plotted by Python module matplotlib, and the lifetimes were plotted in histogram and fitted with a simple exponential decay function by Origin 8.5 to obtain the rate constants. The relative free energy, $\Delta\Delta G$ was measured using the equation $\Delta\Delta G_{ab} = -RT \ln(P_b/P_a)$, where $R$, $T$, $P_a$ and $P_b$ are the gas constant, absolute temperature, population of the observed state b and population of the control state a, respectively (*Manz et al., 2017*).

## In vitro transcription termination assay

The single-round transcription termination assay was carried out by PLOR (*Chien et al., 2023*). The detailed procedure and reagents are listed in *Supplementary file 1,tables S8–16*. 0~ 20mM guanidine and 6 mM $Mg^{2+}$ were added at step 8 in *Supplementary file 1, table S8* to measure the effect of guanidine on transcription read-through of the guanidine-IV riboswitch. The products collected at the last step were loaded to 12% denaturing PAGE to compare the transcription read-through efficiencies at different conditions.

## Acknowledgements

The work was supported by the National Natural Science Foundation of China [grant no. 32071300 and 31872628].

## Additional information

### Funding

| Funder | Grant reference number | Author |
| --- | --- | --- |
| National Natural Science Foundation of China | 32071300 | Yu Liu |
| National Natural Science Foundation of China | 31872628 | Yu Liu |

The funders had no role in study design, data collection and interpretation, or the decision to submit the work for publication.

### Author contributions

Lingzhi Gao, L.G. designed the experiments, collected the data and drafted the manuscript; Dian Chen, D.C. contributed to in vitro data collection; Yu Liu, Y.L. designed the experiments and drafted the manuscript

### Author ORCIDs

Lingzhi Gao (iD) https://orcid.org/0000-0002-4273-7946
Yu Liu (iD) https://orcid.org/0000-0002-4188-2949

Reviewer #1 (Public Review): https://doi.org/10.7554/eLife.94706.3.sa1
Reviewer #2 (Public Review): https://doi.org/10.7554/eLife.94706.3.sa2
Reviewer #3 (Public Review): https://doi.org/10.7554/eLife.94706.3.sa3
Author response https://doi.org/10.7554/eLife.94706.3.sa4

## Additional files

### Supplementary files

• Supplementary file 1. Information of DNA/RNA sequences and reagent usages for PLOR reactions.
• MDAR checklist

### Data availability

All data generated or analysed during this study are included in the manuscript and supporting files.

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
