## [Editor Report · eLife assessment]

This study presents **valuable** findings on the ligand- and ion-dependent structural dynamics of a transcriptional riboswitch. The single-molecule data presented are **solid** and prompts intriguing hypotheses and models, which will undoubtedly stimulate future structural analyses. These findings are of considerable interest to biochemists and biophysicists engaged in the study of RNA structure and riboswitch mechanisms.

---

## [Referee Report · Reviewer #1 (Public Review)]

Summary:

This work presents an in-depth characterization of the factors that influence the structural dynamics of the Clostridium botulinum guanidine-IV riboswitch (riboG). Using a single-molecule FRET, the authors demonstrate that riboG undergoes ligand and Mg2+ dependent conformational changes consistent with dynamic formation of a kissing loop (KL) in the aptamer domain. Formation of the KL is attenuated by Mg2+ and Gua+ ligand at physiological concentrations as well as the length of the RNA. Interestingly, the KL is most stable in the context of just the aptamer domain compared to longer RNAs capable of forming the terminator stem. To attenuate transcription, binding of Gua+ and formation of the KL must occur rapidly after transcription of the aptamer domain but before transcription of the rest of the terminator stem.

Strengths:

(1) Single molecule FRET microscopy is well suited to unveil the conformational dynamics of KL formation and the authors provide a wealth of data to examine the effect of the ligand and ions on riboswitch dynamics. The addition of complementary transcriptional readthrough assays provides further support the author's proposed model of how the riboswitch dynamics contribute to function.

(2) The single-molecule data strongly support that the effect of Gua+ ligand and Mg2+ influence the RNA structure differently for varying lengths of the RNA. The authors also demonstrate that this is specific for Mg2+ as Na+ and K+ ions have little effect.

(3) The PLOR method utilized is clever and well adapted for both dual labeling of RNAs and examining RNA at various lengths to mimic co-transcriptional folding. Using PLOR, they demonstrate that a change in the structural dynamics and ligand binding can occur after extension of the RNA transcript by a single nucleotide. Such a tight window of regulation has intriguing implications for kinetically controlled riboswitches.

(4) In the revised version, the authors utilized multiple destabilizing and compensatory mutations to strengthen their structural interpretation of the KL structure and dynamics and cementing their conclusions.

---

## [Referee Report · Reviewer #2 (Public Review)]

Summary:

Gao et al., used single-molecule FRET and step-wise transcription methods to study the conformations of the recently reported guanidine-IV class of bacterial riboswitches that upregulate transcription in the presence of elevated guanidine. Using three riboswitch lengths, the authors analyzed the distributions and transitions between different conformers in response to different Mg2+ and guanidine concentrations. These data led to a three-state kinetic model for the structural switching of this novel class of riboswitches whose structures remain unavailable. Using the PLOR method that the authors previously invented, they further examined the conformations, ligand responses, and gene-regulatory outcomes at discrete transcript lengths along the path of vectorial transcription. These analyses uncover that the riboswitch exhibits differential sensitivity to ligand-induced conformational switching at different steps of transcription, and identify a short window where the regulatory outcome is most sensitive to ligand binding.

Strengths:

Dual internal labeling of long RNA transcripts remains technically very challenging, but essential for smFRET analyses of RNA conformations. The authors should be commended for achieving very highly quality and purity in their labelled RNA samples. The data are extensive, robust, thorough, and meticulously controlled. The interpretations are logical and conservative. The writing is reasonably clear and illustrations are of high quality. The findings are significant because the paradigm uncovered here for this relatively simple riboswitch class is likely also employed in numerous other kinetically regulated riboswitches. The ability to quantitatively assess RNA conformations and ligand responses at multiple discrete points along the path towards the full transcript provides a rare and powerful glimpse into co-transcriptional RNA folding, ligand-binding, and conformational switching.

Weaknesses:

The use of T7 RNA polymerase instead of a near cognate bacterial RNA polymerase in the termination/antitermination assays is a significant caveat. It is understandable as T7 RNA polymerase is much more robust than its bacterial counterparts, which probably will not survive the extensive washes required by the PLOR method. The major conclusions should still hold, as the RNA conformations are probed by smFRET at static, halted complexes instead of on the fly. However, potential effects of the cognate RNA polymerase cannot be discerned here, including transcriptional rates, pausing, and interactions between the nascent transcript and the RNA exit channel, if any. The authors should refrain from discussing potential effects from the DNA template or the T7 RNA polymerase, as these elements are not cognate with the riboswitch under study.

---

## [Referee Report · Reviewer #3 (Public Review)]

Summary:

In this article, Gao et. al. uses single-molecule FRET (smFRET) and position-specific labelling of RNA (PLOR) to dissect the folding and behavioral ligand sensing of the Guanidine-IV riboswitch in the presence and absence of the ligand guanidine and the cation Mg2+. Results provided valuable information on the mechanistic aspects of the riboswitch, including the confirmation on the kissing loop present in the structure as essential for folding and riboswitch activity. Co-transcriptional investigations of the system provided key information on the ligand-sensing behavior and ligand-binding window of the riboswitch. A plausible folding model of the Guanidine-IV riboswitch was proposed as a final result. The evidence presented here sheds additional light into the mode of action of transcriptional riboswitches.

Strengths:

The investigations were very thorough, providing data that supports the conclusions. The use of smFRET and PLOR to investigate RNA folding has been shown to be a valuable tool to the understand of folding and behavior properties of these structured RNA molecules. The co-transcriptional analysis brought important information on how the riboswitch works, including the ligand-sensing and the binding window that promotes the structural switch. The fact that investigations were done with the aptamer domain, aptamer domain + terminator/anti-terminator region, and the full length riboswitch were essential to inform how each domain contributes to the final structural state if in the presence of the ligand and Mg2+.

Weaknesses:

The system has its own flaws when comparing to physiological conditions. The RNA polymerase used (the study uses T7 RNA polymerase) is different from the bacterial RNA polymerase, not only on complexity, but also in transcriptional speed, that can direct interfere with folding and ligand-sensing. Additionally, rNTPs concentrations were much lower than physiological concentrations during transcription, likely causing a change in the polymerase transcriptional speed. These important aspects and how they could interfere with results are important to be addressed to the broad audience. Another point of consideration to be aware is that the bulky fluorophores attached to the nucleotides can interfere with folding to some extent.

---

## [Author Response]

The following is the authors’ response to the original reviews.

**Public Reviews:**

**Reviewer #1 (Public Review):**
Summary:This work presents an in-depth characterization of the factors that influence the structural dynamics of the Clostridium botulinum guanidine-IV riboswitch (riboG). Using a single-molecule FRET, the authors demonstrate that riboG undergoes ligand and Mg2+ dependent conformational changes consistent with the dynamic formation of a kissing loop (KL) in the aptamer domain. Formation of the KL is attenuated by Mg2+ and Gua+ ligand at physiological concentrations as well as the length of the RNA. Interestingly, the KL is most stable in the context of just the aptamer domain compared to longer RNAs capable of forming the terminator stem. To attenuate transcription, binding of Gua+ and formation of the KL must occur rapidly after transcription of the aptamer domain but before transcription of the rest of the terminator stem.Strengths:(1) Single-molecule FRET microscopy is well suited to unveil the conformational dynamics of KL formation and the authors provide a wealth of data to examine the effect of the ligand and ions on riboswitch dynamics. The addition of complementary transcriptional readthrough assays provides further support for the author's proposed model of how the riboswitch dynamics contribute to function.(2) The single-molecule data strongly support that the effect of Gua+ ligand and Mg2+ influence the RNA structure differently for varying lengths of the RNA. The authors also demonstrate that this is specific for Mg2+ as Na+ and K+ ions have little effect.(3) The PLOR method utilized is clever and well adapted for both dual labeling of RNAs and examining RNA at various lengths to mimic co-transcriptional folding. Using PLOR, they demonstrate that a change in the structural dynamics and ligand binding can occur after the extension of the RNA transcript by a single nucleotide. Such a tight window of regulation has intriguing implications for kinetically controlled riboswitches.Weaknesses:(1) The authors use only one mutant to confirm that their FRET signal indicates the formation of the KL. Importantly, this mutation does not involve the nucleotides that are part of the KL interaction. It would be more convincing if the authors used mutations in both strands of the KL and performed compensatory mutations that restore base pairing. Experiments like this would solidify the structural interpretation of the work, particularly in the context of the full-length riboG RNA or in the cotranscriptional mimic experiments, which appear to have more conformational heterogeneity.

We thank the reviewer for describing our work “in-depth characterization” of riboG. We agree with the reviewer and we have added two more mutants, G71C and U72C with the mutations located at the KL (Figure 2– figure supplement 8A, 8B, 9A, 9B, Figure 3– figure supplement 6A, 6B, 7A, 7B, and Figure 4– figure supplement 6A, 6B, 7A, 7B). Furthermore, we have performed compensatory mutations, C30G-G71C and A29G-U72C that restore base pairing in the KL (Figure 2– figure supplement 8C, 8D, 9C, 9D, Figure 3– figure supplement 6C, 6D, 7C, 7D, and Figure 4– figure supplement 6C, 6D, 7C, 7D). We added the experimental results in the revised manuscript accordingly as “The highly conserved nucleotides surrounding the KL are crucial for its formation (Lenkeit et al., 2020). To test our hypothesis that the state with EFRET ~ 0.8 corresponds to the conformation with the KL, we preformed smFRET analysis on several mutations at these crucial nucleotides (Figure 2– figure supplement 8–10). Consistent with our expectations, the peaks with EFRET ~ 0.8 was significantly diminished in the riboG-G71C mutant, which features a single nucleotide mutation at site 71 (with 97% nucleotide conservation) in the KL (Figure 2– figure supplement 8A and 8B). It is worth noting that the C30G and G71C mutant, which were initially expected to restore a base pair in the KL, did not successfully bring about the anticipated peak of EFRET ~ 0.8 (Figure 2– figure supplement 8C and 8D). On the other hand, the riboG-U72C mutant exhibited a lower proportion at the state with EFRET ~ 0.8 than riboG-apt. However, the A29G and U72C mutations restored a base pair in the KL, as well as the formation of the KL (Figure 2– figure supplement 9). Furthermore, our investigation revealed that the G77C mutant, involving a single nucleotide mutation at a highly conversed site, 77 (with 97% nucleotide conservation), also hindered the formation of the KL (Figure 2– figure supplement 10). This finding aligns with previous research (Lenkeit et al., 2020) and the predicted second structure of G77C mutation by Mfold (Zuker, 2003)” (page 7), “In contrast to riboG-term, both its G71C and C30G-G71C mutants displayed a reduced proportion of the state with EFRET ~ 0.8. Remarkably, the fractions of EFRET ~ 0.8 remained unaffected by the addition of 1.0 mM Gua+ in these mutants. Distinct from riboG-term, no structural transitions between states were observed in the two mutants (Figure 3– figure supplement 6). Regarding the U72C mutant of riboG-term, the mutation at the site 72 had a reduced impact on the KL conformation in the presence of 1.0 mM Gua+ and 2.0 mM Mg2+. However, the increased proportion of EFRET ~ 0.8 in the A29G-U72C mutant of riboG-term suggests that these mutations can restore the base-pairing between sites 29 and 72, as well as facilitate the formation of the KL (Figure 3– figure supplement 7)” (page 8), and “Upon comparing the G71C and C30G-G71C mutants of the full-length riboG with their wild-type counterpart, it was observed that the wild-type adopted higher proportions of the state with EFRET ~ 0.8 (Figure 4– figure supplement 6). Regarding the U72C and A29G-U72C mutants of the full-length riboG, their behaviors with regards to the peak with EFRET ~ 0.8 were similar to that of their counterparts in riboG-term (Figure 4– figure supplement 7)” (page 9).

(2) The existence of the pre-folded state (intermediate FRET ~0.5) is not well supported in their data and could be explained by an acquisition artifact. The dwell times are very short often only a single frame indicating that there could be a very fast transition (< 0.1s) from low to high FRET that averages to a FRET efficiency of 0.5. To firmly demonstrate that this intermediate FRET state is metastable and not an artifact, the authors need to perform measurements with a faster frame rate and demonstrate that the state is still present.

We thank the reviewer for the great comment. We added smFRET experiments at higher time resolution, 20 ms, as well as lower time resolution (Figure 2– figure supplement 3). Based on our experimental results, the intermediate state (EFRET ~0.5) exists at the smFRET collected at 20 ms, 100 ms and 200 ms.

(3) The PLOR method employs a non-biologically relevant polymerase (T7 RNAP) to mimic transcription elongation and folding near the elongation complex. T7 RNAP has a shorter exit channel than bacterial RNAPs and therefore, folding in the exit channel may be different between different RNAPs. Additionally, the nascent RNA may interact with bacterial RNAP differently. For these reasons, it is not clear how well the dynamics observed in the T7 ECs recapitulate riboswitch folding dynamics in bacterial ECs where they would occur in nature.

We thank the reviewer for the comment. We agree with the reviewer that the bacterial and T7 RNAPs may behave differently due to their differences in transcriptional speed, dynamics, interactions, and so on. And we added a statement in the Discussion as “It is worth noting that the RNAP utilized in our study is T7 RNAP, which exhibits distinct characteristics compared to bacterial RNAP in terms of transcriptional speed, dynamics, and interactions. However, Xue et al. have reported similarities between T7 and *E. coli* RNAP in the folding of nascent RNA. Additionally, Lou and Woodson have provided valuable insights into the co-transcriptional folding of the glmS ribozyme using T7 RNAP (Xue et al., 2023; Lou & Woodson, 2024)” (page 13–14).

**Reviewer #2 (Public Review):**
Summary:Gao et al. used single-molecule FRET and step-wise transcription methods to study the conformations of the recently reported guanidine-IV class of bacterial riboswitches that upregulate transcription in the presence of elevated guanidine. Using three riboswitch lengths, the authors analyzed the distributions and transitions between different conformers in response to different Mg2+ and guanidine concentrations. These data led to a three-state kinetic model for the structural switching of this novel class of riboswitches whose structures remain unavailable. Using the PLOR method that the authors previously invented, they further examined the conformations, ligand responses, and gene-regulatory outcomes at discrete transcript lengths along the path of vectorial transcription. These analyses uncover that the riboswitch exhibits differential sensitivity to ligand-induced conformational switching at different steps of transcription, and identify a short window where the regulatory outcome is most sensitive to ligand binding.Strengths:Dual internal labeling of long RNA transcripts remains technically very challenging but essential for smFRET analyses of RNA conformations. The authors should be commended for achieving very high quality and purity in their labelled RNA samples. The data are extensive, robust, thorough, and meticulously controlled. The interpretations are logical and conservative. The writing is reasonably clear and the illustrations are of high quality. The findings are significant because the paradigm uncovered here for this relatively simple riboswitch class is likely also employed in numerous other kinetically regulated riboswitches. The ability to quantitatively assess RNA conformations and ligand responses at multiple discrete points along the path towards the full transcript provides a rare and powerful glimpse into cotranscriptional RNA folding, ligand-binding, and conformational switching.Weaknesses:The use of T7 RNA polymerase instead of a near-cognate bacterial RNA polymerase in the termination/antitermination assays is a significant caveat. It is understandable as T7 RNA polymerase is much more robust than its bacterial counterparts, which probably will not survive the extensive washes required by the PLOR method. The major conclusions should still hold, as the RNA conformations are probed by smFRET at static, halted complexes instead of on the fly. However, potential effects of the cognate RNA polymerase cannot be discerned here, including transcriptional rates, pausing, and interactions between the nascent transcript and the RNA exit channel, if any. The authors should refrain from discussing potential effects from the DNA template or the T7 RNA polymerase, as these elements are not cognate with the riboswitch under study.

We thank the reviewer for describing our work “The data are extensive, robust, thorough, and meticulously controlled. The interpretations are logical and conservative. The writing is reasonably clear and the illustrations are of high quality”. We agree with the reviewer that the bacterial and T7 RNAPs may behave differently due to their differences in transcriptional speed, dynamics, interactions, and so on. And we added a statement in the Discussion as “It is worth noting that the RNAP utilized in our study is T7 RNAP, which exhibits distinct characteristics compared to bacterial RNAP in terms of transcriptional speed, dynamics, and interactions. However, Xue et al. have reported similarities between T7 and *E. coli* RNAP in the folding of nascent RNA. Additionally, Lou and Woodson have provided valuable insights into the co-transcriptional folding of the glmS ribozyme using T7 RNAP (Xue et al., 2023; Lou & Woodson, 2024)” (page 14).

**Reviewer #3 (Public Review):**
Summary:In this article, Gao et. al. uses single-molecule FRET (smFRET) and position-specific labelling of RNA (PLOR) to dissect the folding and behavioral ligand sensing of the Guanidine-IV riboswitch in the presence and absence of the ligand guanidine and the cation Mg2+. The results provided valuable information on the mechanistic aspects of the riboswitch, including the confirmation of the kissing loop present in the structure as essential for folding and riboswitch activity. Co-transcriptional investigations of the system provided key information on the ligand-sensing behavior and ligandbinding window of the riboswitch. A plausible folding model of the Guanidine-IV riboswitch was proposed as a final result. The evidence presented here sheds additional light on the mode of action of transcriptional riboswitches.Strengths:The investigations were very thorough, providing data that supports the conclusions. The use of smFRET and PLOR to investigate RNA folding has been shown to be a valuable tool for the understanding of folding and behavior properties of these structured RNA molecules. The co-transcriptional analysis brought important information on how the riboswitch works, including the ligand-sensing and the binding window that promotes the structural switch. The fact that investigations were done with the aptamer domain, aptamer domain + terminator/anti-terminator region, and the full-length riboswitch were essential to inform how each domain contributes to the final structural state if in the presence of the ligand and Mg2+.Weaknesses:The system has its own flaws when compared to physiological conditions. The RNA polymerase used (the study uses T7 RNA polymerase) is different from the bacterial RNA polymerase, not only in complexity, but also in transcriptional speed, which can directly interfere with folding and ligand-sensing. Additionally, rNTPs concentrations were much lower than physiological concentrations during transcription, likely causing a change in the polymerase transcriptional speed. These important aspects and how they could interfere with results are important to be addressed to the broad audience. Another point of consideration to be aware of is that the bulky fluorophores attached to the nucleotides can interfere with folding to some extent.

We thank the reviewer for describing our work as “The investigations were very thorough, providing data that supports the conclusions”. We agree with the reviewer that the bacterial and T7 RNAPs may behave differently due to their differences in transcriptional speed, dynamics, interactions, and so on. And we added a statement in the Discussion as “It is worth noting that the RNAP utilized in our study is T7 RNAP, which exhibits distinct characteristics compared to bacterial RNAP in terms of transcriptional speed, dynamics, and interactions. However, Xue et al. have reported similarities between T7 and *E. coli* RNAP in the folding of nascent RNA. Additionally, Lou and Woodson have provided valuable insights into the cotranscriptional folding of the glmS ribozyme using T7 RNAP (Xue et al., 2023; Lou & Woodson, 2024)” (page 14). And we also agree with the reviewer that the lower NTP may affect the transcriptional speed. Regarding the fluorophores, we purposely placed them away from the KL to avoid their influence on the formation of the KL.

**Reviewer #1 (Recommendations For The Authors):**
Related to weakness 1- The authors cite a paper that investigated mutations in the KL duplex but do not include these mutations in their analysis. It is unclear why the authors chose the G77C mutation and not the other mutants previously tested. Can the authors explain their choice of mutation in detail in the text? I also did not see the proposed secondary structure for the G77C mutant shown in Figure 2 -supp 3A in the cited paper, is this a predicted structure? Please explain how this structure was determined.

We thank the reviewer for the comment. The reason we chosen the G77C mutation is based on previous report that G77C can disturb the formation of the KL, as we stated in the manuscript as “Furthermore, our investigation revealed that the G77C mutant, involving a single nucleotide mutation at a highly conversed site, 77 (with 97% nucleotide conservation), also hindered the formation of the KL (Figure 2– figure supplement 10). This finding aligns with previous research (Lenkeit et al., 2020) and the predicted second structure of G77C mutation by Mfold (Zuker, 2003)” (page 7). And the secondary structure for the G77C mutant was predicted by Mfold, which as cited in the manuscript and added in the reference list as “Zuker, M. (2003). Mfold web server for nucleic acid folding and hybridization prediction. Nucleic Acids Research, 31(13), 3406-3415”.

- It is not clear to me that the structural interpretation of their FRET states is correct and that the FRET signal reports on the base pairing of the KL in only the high FRET state. The authors should perform experiments with additional mutations in the KL duplex to confirm that their construct reports on KL duplex formation alone and not other structural dynamics.

We thank the reviewer for the comment. We have included additional mutations to establish a connection between the high-FRET state to the formation of the KL. The results have been added to the manuscript as “The highly conserved nucleotides surrounding the KL are crucial for its formation (Lenkeit et al., 2020). To test our hypothesis that the state with EFRET ~ 0.8 corresponds to the conformation with the KL, we preformed smFRET analysis on several mutations at these crucial nucleotides (Figure 2– figure supplement 8–10). Consistent with our expectations, the peaks with EFRET ~ 0.8 was significantly diminished in the riboG-G71C mutant, which features a single nucleotide mutation at site 71 (with 97% nucleotide conservation) in the KL (Figure 2– figure supplement 8A and 8B). It is worth noting that the C30G and G71C mutant, which were initially expected to restore a base pair in the KL, did not successfully bring about the anticipated peak of EFRET ~ 0.8 (Figure 2– figure supplement 8C and 8D). On the other hand, the riboG-U72C mutant exhibited a lower proportion at the state with EFRET ~ 0.8 than riboG-apt. However, the A29G and U72C mutations restored a base pair in the KL, as well as the formation of the KL (Figure 2– figure supplement 9). Furthermore, our investigation revealed that the G77C mutant, involving a single nucleotide mutation at a highly conversed site, 77 (with 97% nucleotide conservation), also hindered the formation of the KL (Figure 2– figure supplement 10). This finding aligns with previous research (Lenkeit et al., 2020) and the predicted second structure of G77C mutation by Mfold (Zuker, 2003)” (page 7), “In contrast to riboG-term, both its G71C and C30G-G71C mutants displayed a reduced proportion of the state with EFRET ~ 0.8. Remarkably, the fractions of EFRET ~ 0.8 remained unaffected by the addition of 1.0 mM Gua+ in these mutants. Distinct from riboG-term, no structural transitions between states were observed in the two mutants (Figure 3– figure supplement 6). Regarding the U72C mutant of riboG-term, the mutation at the site 72 had a reduced impact on the KL conformation in the presence of 1.0 mM Gua+ and 2.0 mM Mg2+. However, the increased proportion of EFRET ~ 0.8 in the A29G-U72C mutant of riboG-term suggests that these mutations can restore the base-pairing between sites 29 and 72, as well as facilitate the formation of the KL (Figure 3– figure supplement 7)” (page 8), and “Upon comparing the G71C and C30G-G71C mutants of the full-length riboG with their wild-type counterpart, it was observed that the wild-type adopted higher proportions of the state with EFRET ~ 0.8 (Figure 4– figure supplement 6). Regarding the U72C and A29G-U72C mutants of the full-length riboG, their behaviors with regards to the peak with EFRET ~ 0.8 were similar to that of their counterparts in riboG-term (Figure 4– figure supplement 7)” (page 9).

- For the full-length riboG-136 (Cy3Cy5 riboG in Figure 4), the authors have clearly defined peaks at 0.6 and 0.4. However, the authors do not explain their structural interpretation of these states. Do the authors believe that the KL is forming in these states? It would be helpful to have data on mutations in the KL in the context of the full-length riboG to better understand the structural transitions of these intermediate states.

Based on our mutation studies, we proposed that the peak with EFRET ~0.8 corresponds to the conformation with the KL, while the states with EFRET ~0.4 and 0.6 are the states without a stable KL.

Related to weakness 2:- For the riboG-apt and riboG-term RNAs, the proposed intermediate FRET state (EFRET = 0.5) is poorly fit by a Gaussian and the dwell times in the state are almost entirely single-frame dwells. It is likely that this state is the result of a camera blurring artifact, in which RNAs undergo a FRET transition between two frames giving an apparent FRET efficiency which is between that of the two transitioning states. This artifact arises when the average dwell times of the true states (Elow and Ehigh) are comparable to the frame duration (within a factor of ~5-10; see https://doi.org/10.1021/acs.jpcb.1c01036). To confirm the presence of the intermediate state, the authors should perform at least a few experiments with higher time resolution to support the existence of the 0.5 state with a lifetime of 0.1 s. Alternatively, the data should be refit to a two-state HMM and the authors could explain in the text that the density in the FRET histogram between the two states is likely due to transitions that are faster than the time resolution of the experiment.

We thank the reviewer for the great comment. Taking the suggestion into consideration, we performed smFRET experiments with a higher time resolution of 20 ms. As a result, we still detected the intermediate state, supporting that it is not an artifact. The new data has been included in the revised manuscript (Figure 2-figure supplement 3).

Related to weakness 3:- The authors depict the polymerase footprint differently in some of the figures and it is unclear if this is part of their model. Is the cartoon RNAP supposed to indicate the RNA:DNA hybrid or the footprint of T7 RNAP on the RNA? For example, in Figure 8a there are 8 nts (left) and 9 nts (right) covered by RNAP, and only 6nts in Figure 6 - supp 2A. This is particularly misleading for the EC-87 and EC-88 in Figure 6 - supp 2, where it is likely that this stem is not formed at all and the KL strand is single-stranded. The authors should clarify and at least indicate in the figure legend if the RNAP cartoon is part of the model or only a representation.

We thank the reviewer for bringing the issues to our attention. Due to space limitations, we chose to represent the polymerase footprint differently in Figure 8. However, we have included the statement “DNA templates from EC-87 to EC-105 are not displayed in the model” in the legend of Figure 8 to avoid the confusion.

Moreover, we have corrected the error of 6 nts Figure 6-supplement figure 2.

- With a correct 9 bp RNA:DNA hybrid, the EC-88 construct would not be able to form the top part of the P2 stem and the second half of the KL RNA would be single-stranded. In this case, an interaction between the KL nucleotides would resemble a pseudoknot and not a kissing loop interaction. Can the authors explain if this could explain the heterogeneity they observe in the EC-88 construct compared to the riboGapt RNA?

Thank the reviewer for the comment. We have added the statement in the revised manuscript as “The T7 RNA polymerase (RNAP) sequestered about 8 nt of the nascent RNA, preventing the EC-88 construct from forming the P2 stem (Durniak et al., 2008; Huang & Sousa, 2000; Lubkowska et al., 2011; Tahirov et al., 2002; Wang et al., 2022; Yin & Steitz, 2002). Consequently, a pseudoknot structure potentially formed instead of the expected KL. This distinction may account for the observed heterogeneity between EC-88 and riboG-apt” (page 11).

Other comments:(1) It appears that the FRET histograms in the PLOR experiments (Figure 6 and related figures) only show the fits presumably to highlight the overlays. However, this makes it impossible to determine the goodness of the fit. The authors should instead show the outline of the raw histogram with the fit, or at least show the raw histograms with fits in the supplement.

We have replaced Figure 6- figure supplements 2-4 to enhance the clarity of the raw and fitted smFRET histograms.

(2) The authors should consider including a concluding paragraph to put the results into a larger context. How does the kinetic window compare to other transcriptional riboswitches? Would the authors comment on how the transcription speed compares to the kinetics for the formation of the KL?

We thank the reviewer for the comment. We have added the comparison of riboG to other transcription riboswitches to the manuscript as “Nevertheless, the ligand-sensitive windows of riboswitches during transcription vary. In a study conducted by Helmling et al. using NMR spectroscopy, they proposed a broad transcriptional window for deoxyguanosine-sensing riboswitches, whereby the ligand binding capability gradually diminishes over several nucleotide lengths (Helmling et al., 2017). However, more recent research by Binas et al. and Landgraf et al. on riboswitches sensing ZMP, c-di-GMP, and c-GAMP revealed a narrow window with a sharp transition in binding capability, even with transcript lengths differing by only one or three nucleotides (Binas et al., 2020; Landgraf et al., 2022). In line with the findings for the c-GAMP-sensing riboswitch, our study on the guanidine-IV riboswitch also demonstrated a sharp transition in binding capability with just a single nucleotide extension” (page 14).

We appreciate the reviewer’s comment in comparing the transcription speed to the kinetics of the KL formation. However, we must acknowledge that we have limited kinetic data in this study to confidently make such a comparison.

(3) Cy3Cy5 RiboG is a confusing name because it implies that the others are not also Cy3Cy5 labeled. The authors should consider changing the names and being consistent throughout. I suggest full-length riboG or riboG-136.

We have changed “Cy3Cy5 riboG” to “Cy3Cy5-full-length riboG” (pages 15 and 16).

(4) The transcriptional readthrough experiment should be explained when first mentioned in line 109.

We have added the citation (Chien et al., 2023) of the transcriptional readthrough experiment to the manuscript as “we noted that the transcriptional read-through of the guanidine-IV riboswitch during the single-round PLOR reaction was sensitive to Gua+, exhibiting an apparent EC50 value of 68.7 μM 7.3 μM (Figure 1D) (Chien et al., 2023)” (page 5).

(5) Kd values in text should have uncertainties, and the way these uncertainties are obtained should be explained.

We have added the uncertainties of Kd values in the revised manuscript (page 6) and the legend of Figure 2-supplement 6 as “The percentages of the folded state (EFRET ~ 0.8) of Cy3Cy5-riboG-apt were plotted with the concentrations of Gua+ at 0.5 mM Mg2+, with an apparent Kd of 286.0 μM 18.1 μM in three independent experiments”.

(6) The authors mention "strategies" on line 306, but it is unclear what they are referring to. Are the strategies referring to the constructs (EC-87, etc) or Steps 1-8 in the supplemental figure? Please clarify.

We have clarified the confusion by adding “The detailed procedures of strategies 1-8 were shown in Figure 7–figure supplement 1” to the manuscript (page 12).

(7) What are the fraction of dynamic traces versus static traces in the cases for the full-length riboG? This would help depict the structural heterogeneity in the population.

We have added the fractions of dynamic single-molecule traces of the full-length riboG to Figure 4-supplements 1-5.

(8) The labels in Figure 4 (A-E) don't match the caption (A-H).

We have corrected the error.

(9) The coloring of the RNA strands in Figure 4A should be explained in the figure legend. It could be interpreted as multiple strands annealed instead of a continuous strand.

We have revised the legend of Figure 4A by adding “The full-length riboG contains the aptamer domain (black), terminator (red) and the extended sequence (blue). Cy3 and Cy5 are shown by green and red sparkles, respectively”.

(10) Reported quantities and uncertainties should have the same number of decimal places. In many places, the uncertainties likely have too many significant figures, for example, in Figure 5 and related figures.

We have corrected the significant figures of the uncertainties.

(11) In Figure 5, A and B should have the same vertical scale to facilitate comparison.

We have adjusted Figure 5A to match the vertical scale of Figure 5B in the revised manuscript.

(12) In Figure 5C-D, the construct from which those trajectories come should be indicated in the legend.

We have added the construct to the legend of Figures 5C and D.

(13) In Figure 6J, the splines between data points are confusing and can be misleading. They suggest that the data has been fit to a model, but I am not sure if it represents a model. The data points should be colored instead and lines removed.

We thank the reviewer for the comment. We have changed Figure 6J by coloring the data points and removing the lines to avoid confusion.

(14) Line 330 mentions a P2 structure in Figure 8, but there is no such label in Figure. Please clarify.

We thank the reviewer for the comment and have added P2 to Figure 8.

**Reviewer #2 (Recommendations For The Authors):**
(1) Figure 1B. The authors don't seem to address the role of the blue stem-loop following Stems 1 and 2. Is this element needed at all for gene regulation? Does it impact the conformations or folding of the preceding Stems 1 and 2? It seems feasible to disrupt the stem and see whether there is an impact on riboswitch function.

We thank the reviewer for the comment. The presence of the sequence which formed blue stem-loop indicates the formation of an anti-terminator conformation in riboG during transcription. Our smFRET data shows that the inclusion of the stem-loop sequence induces additional peaks in the full-length riboG compared to the riboGterm. This indicates that the stem-loop influences the folding of the kissing loop (KL) and potentially also affects the stems 1 and 2.

(2) Figure 7 supplement 1, C &D. Maybe I am missing something, but it seems to me in reaction #8 (EC-105, last two lanes), the readthrough percentage is close to 50% based on the gel but plotted in D as 20%. Further, there is a strong effect of guanidine in reaction #8 but that is not reflected in the quantitation in panel D.

We thank the reviewer for the comment. The observed discrepancy between reaction 8 in (C) and (D) is from the differential handling of the crude product at the last step (step 17) in gel loading for (C), contrasted with the combination of crude products from steps 16 and 17 to calculate the read-through percentage in (D). We have corrected the discrepancy by replacing Figure 7-Supplement figure 1C (now Figure 7C), and revised the legend to include the following clarification: “Taking into consideration that the 17 step-PLOR reaction exhibited a pause within the terminator region, resulting in a significant amount of terminated product at step 16, crude products from steps 16 and 17 were collected for (C) and (D) of the 17 step-PLOR reaction (Lanes 15 and 16 in C)”.

(3) Figure 7C is a control that shows the quality of the elongation complexes, which probably should be in the supplement. Instead, in Figure 7 supplement 1, panels C and D are actual experiments and could be moved into the main figure.

We thank the reviewer for the comment. We made the adjustment.

(4) Figure S7D. I would suggest not labelling the RNA polymerase halt/stoppage sites due to NTP deprivation as "pausing sites" because transcriptional pausing has previously been defined as natural sites where the RNA polymerase transiently halts itself, but not due to the lack of the next NTPs. In this case, the elongating complexes were artificially halted, which is technically not "pausing", as it will not restart/resume on its own without intervention.

We have changed the “pausing” to “halting”.

(5) Figure 7 is titled "In vitro transcriptional performance of riboG." But the data is actually not about the performance of the riboswitch, or how well it functions. I would suggest the authors revise the title. This is mostly about the observed sensitivity window of the riboswitch to ligand-mediated conformational switching.

We have changed the title of Figure 7 to “Ligand-mediated conformational switching of riboG during transcription”.

(6) Figure 7A, the illustration gives the visual impression that there are multiple RNA polymerases on the same DNA template, which is not the case.

We have revised Figure 7A by adding arrows between RNA polymerases to illustrate the movement of a single RNAP, rather than multiple RNAP on the same template.

(7) It could be informative to compare the guanidine-IV riboswitch with the first three classes (I, II, III), to see how their architectures or gene regulatory mechanisms are similar or different.

We thank the reviewer for the comment. We have added the comparison of the guanidine-IV riboswitch to other three guanidine riboswitches to the manuscript as “The guanidine-IV riboswitch exhibits similarities to the guanidine-I riboswitch in gene regulatory mechanism, functioning as a transcriptional riboswitch. Structurally, it resembles the guanidine-II riboswitch through the formation of loop-loop interactions upon binding to guanidine (Battaglia & Ke, 2018; L. Huang et al., 2017; Lin Huang et al., 2017; Lenkeit et al., 2020; Nelson et al., 2017; Reiss & Strobel, 2017; Salvail et al., 2020)” (page 12).

**Reviewer #3 (Recommendations For The Authors):**
In addition to the public review items, I provide the following recommendations:(1) As a second language speaker, I understand that writing a compelling and concise story may be hard, and we tend to write more than needed or more repetitively. That being said, I do think that the writing could be improved to make it more concise, clear, and avoid repetitions.

We thank the reviewer for the comment. We re-wrote the abstract and some sentences in the manuscript.

(2) In the abstract, instead of saying that "...This lack of understanding has impeded the application of this riboswitch", which makes the statement too strong, perhaps, stating something along the lines of "this understanding would assist the application of this riboswitch", would be a better fit.

We have re-wrote the abstract, and revised the sentence.

(3) Methods should state which RNA polymerase was used. PLOR uses T7 RNA pol, so I assume it was the same.

We have added the statement “T7 RNAP was utilized in the PLOR and in vitro transcription reactions except noted” in the Methods (page 15).

(4) The impact statement says comprehensive structure-function, where perhaps comprehensive folding-function would be more appropriate. We are still missing a lot of structural information about this particular riboswitch.

We agree with the reviewer, and changed “comprehensive structure-function” to “folding-function” in Impact statement (page 2).

(5) Higher Mg2+ concentrations implicated in a lesser extent of the switch of RiboGapt, a sentence talking about it would be useful (how Mg2+ could have promiscuous interaction and interfere with folding).

We have added the role of higher Mg2+ to the manuscript as “However, at a higher concentration of 50.0 mM Mg2+, the proportion of the pre-folded and unfolded conformations were more prevalent at 50.0 mM Mg2+ than at 20.0 mM Mg2+. This suggests that an excess of Mg2+ may promote the pre-folded and even unfolded conformations” (page 6).

(6) In the investigations of RiboG-term and RiboG, seems like that monovalents from the buffer are sufficient to promote secondary structure. A statement commenting on this would benefit the paper and the audience.

We agree with the reviewer and have accordingly revised the manuscript accordingly by adding “This indicates that monovalent ions in the buffer can facilitate the formation of stable guanidine-IV riboswitch” (page 8).

(7) Figure 3. Figure goes to panel E and legend to panel H. G and H colors do not correspond to actual figure colors.

We made the correction.

(8) Figure 4. The same as Figure 3, the panels and figures are divergent.

We made the correction.

(9) During the discussion, stating that the DNA and RNA pol play a role in folding and ligand binding may be excessive. This could be an indirect effect of the transcriptional bubble hindering part of the nascent RNA from folding, which is something intrinsic to any transcription and not specific to this system.

We agree with the reviewer and deleted the statement about the DNA and RNAP play a role in folding and ligand binding.

(10) PLOR is not properly cited. When introduced in the manuscript, please cite the original PLOR paper (Liu et. al. Nature 2015) and additional related papers.

We cited the original PLOR paper (Liu et al, Nature 2015) and the related papers (Liu et al, Nature Protocols 2018). (pages 4 and 15)

(11) The kinetics race of folding and binding could be a little more emphasized in discussion, particularly from the perspective of its physiological importance.

We agree with the reviewer and deleted the kinetics race of folding and binding from the Discussion part.